# AlignCLIP: Enhancing Stable Representations in Vision-Language Pretraining Models through Attention and Prediction Alignment

## Abstract

Stable representations are pivotal in Vision-Language Pretraining (VLP) models serving as the foundation for managing domain shifts and recognizing unseen classes in open-world environments. In this paper, we identify and delve into two primary misalignment problems in VLP models like contrastive language-image pre-training (CLIP): attention misalignment, where the model disproportionately allocates attention to background visual tokens, and predictive category misalignment, indicative of the model's struggle to discern class similarities accurately. Addressing these misalignments is paramount, as they undermine the stability of representations and, consequently, the adaptability and trustworthiness of the model in open-world environments. To counteract these misalignments, we introduce AlignCLIP, a new parameter fine-tuning method. AlignCLIP introduces a novel training objective, the attention alignment loss, to harmonize attention distributions of multi-head attention layers and the correlation between visual tokens and class prompts. Further, AlignCLIP presents semantic label smoothing, aiming to preserve prediction hierarchy based on class similarity derived from textual information. Our empirical studies across varied datasets and out-of-distribution contexts demonstrate AlignCLIP's superior performance in enhancing stable representations and excelling in generalization methodologies, proving its adaptability and stability in scenarios involving domain shifts and unseen classes.

## 1 Introduction

Stable representations are essential in Vision-Language Pretraining (VLP) models, acting as the cornerstone for adaptability in open-world downstream tasks. Firstly, stable representations are fundamental for navigating the myriad of unknown scenarios encountered in open-world environments, such as domain shifts and unseen classes (Taori et al., 2020; Gulrajani & Lopez-Paz, 2021; Miller et al., 2021). The emergence of VLP models has demonstrated the potential of a unified approach in addressing the challenges of open-world environments. VLP models like contrastive language-image pre-training (CLIP) (Radford et al., 2021) have achieved unprecedented success in tasks related to distribution shifts and zero-shot learning datasets (Li et al., 2022a; Singh et al., 2022; Lüddecke & Ecker, 2022), underscoring the importance of stable representations in generalizing acquired knowledge to diverse contexts effectively. However, CLIP sometimes falters when generalizing to data outside their pre-training dataset (Radford et al., 2021). Fine-tuning (Girshick et al., 2014) and prompt learning (Zhou et al., 2022b) can improve performance on downstream tasks but may affect the model's generalization in out-of-distributions (Kornblith et al., 2019; Pham et al., 2021; Wortsman et al., 2022). For instance, fine-tuned CLIP models sometimes underperform zero-shot models on downstream tasks with distribution shifts (Radford et al., 2021; Pham et al., 2021; Wortsman et al., 2022). This underscores the significance of stable representations in ensuring a balance between adaptability and generalization in VLP models.

In this paper, we identify two critical misalignment problems that compromise the stability of representations in CLIP, depicted in Figure 1. The first problem is attention misalignment, a significant impediment to CLIP's generalization on domain shifts (Izmailov et al., 2022) and zero-shot classification (Romera-Paredes & Torr, 2015). The attention misalignment manifests as skewed attention weights in the multi-head attention layers of the image encoder in CLIP, favoring background visual

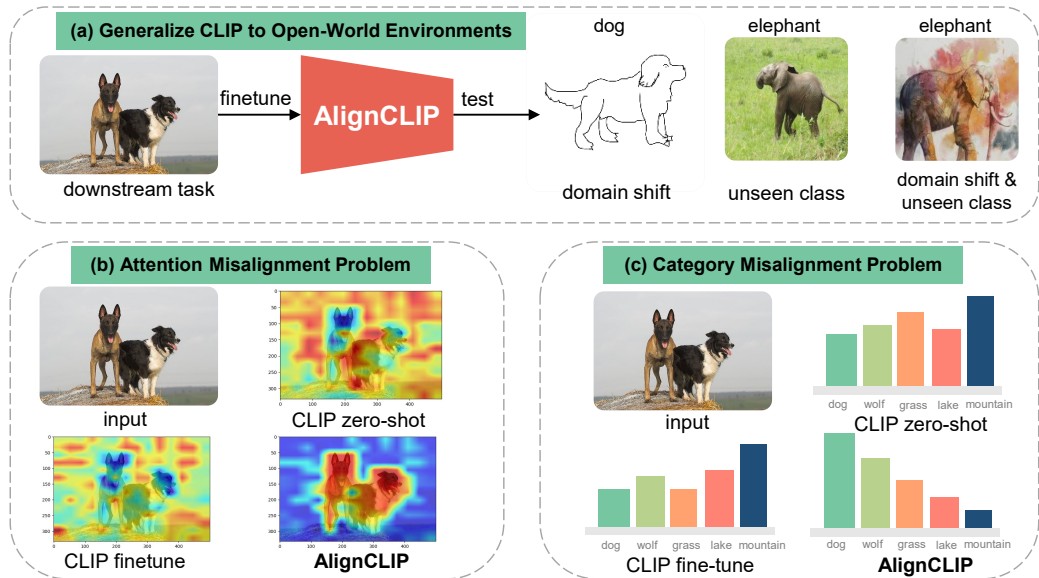

Figure 1: The objective of this paper is to fine-tune pre-trained CLIP models for downstream tasks, with an emphasis on boosting their capacity in open-world scenarios. Images in (b) represent skewed similarity maps, while those in (c) depict the imbalanced distribution of prediction probabilities.

tokens over the primary objects of interest, a finding confirmed by existing interpretability research (Li et al., 2022b; 2023). Further, our research reveals that even thorough traditional fine-tuning of the image encoder's full parameters is insufficient to rectify this problem (Shu et al., 2023). The second problem is predictive category misalignment, leading to models classifying images into unrelated categories and neglecting to choose the most plausible alternative in case errors are unavoidable after fine-tuning. For example, a dog is misclassified as a mountain rather than a more reasonable misclassification like a wolf. This issue arises from visual representations' struggle to discern class similarities, undermining their power to identify unseen classes. Both emphasized misalignments underscore the intrinsic difficulties in enhancing stable representations.

Specifically, we present AlignCLIP, a new method for fine-tuning CLIP models, aimed at enhancing the stability of representations across a spectrum of downstream tasks. AlignCLIP adopts text-prompt-driven fine-tuning that compares the embeddings of images with text-generated embeddings derived from task prompts to refine the visual representations. Based on that, to optimize the attention distribution of multi-head attention layers, AlignCLIP introduces a novel training objective named Attention Alignment Loss (AAL). AAL aligns the attention distribution within multi-head attention layers according to the similarity distribution existing between visual tokens and class-specific text. Based on a diagonal matrix-based D-V structure, AAL customizes the optimization process for both query and key parameters within multi-head attention layers, consequently enhancing representation stability during the fine-tuning process. From the standpoint of model regularization, AlignCLIP introduces Semantic Label Smoothing (SLS). SLS aligns the predicted category distribution with the inherent semantic relationships among categories, as established during language-vision pre-training.

Our core contributions can be summarized as follows:

- We conduct an investigation into a previously under-studied challenge—the ability of CLIP models to generalize to open-world environments, particularly in more complex in-the-wild settings where both domain shift and unseen class problems occur in test data.

- We propose AlignCLIP to handle the two misalignment problems. AlignCLIP innovates through its introduction of a new optimization objective Attention Alignment Loss and a new regularization technique Semantic Label Smoothing. These components are intricately designed to bolster the representation ability on downstream tasks.

- Through exhaustive empirical validation on an array of datasets and across diverse scenarios—including domain shifts, unseen classes, and their co-occurrences—we establish that AlignCLIP markedly supersedes existing techniques in terms of generalization performance.

## 2 RELATED WORK

**Representation Learning in Open-World Environments.** Stable representations are fundamental to the efficacy of machine learning models in open-world environments. For instance, the concept of domain-invariant representation learning is integral to the advancements in deep unsupervised domain adaptation (Han et al., 2022a; Ganin & Lempitsky, 2015; Han et al., 2022b). It is not an isolated case that a plethora of domain generalization methods are designed to learn stable representations through various strategies such as domain alignment (Muandet et al., 2013; Li et al., 2018), causal representation learning (Schölkopf et al., 2021), stable learning (Zhang et al., 2021b), and invariance-based optimization (Liu et al., 2021). In parallel, the detection of unseen categories has given rise to the specialized subfield of zero-shot learning (Pourpanah et al., 2022) and out-of-distribution (OOD) detection (Yang et al., 2021). For example, the foundational premise of both classification-based and density-based OOD detection techniques is the distinguishability of the representations between seen and unseen classes (Hendrycks & Gimpel, 2017; Liang et al., 2017; Gomes et al., 2022; Du et al., 2022; Jiang et al., 2021). These techniques underscore the importance of stable representations in enhancing the model's ability to adapt to new distributions and identify unseen categories effectively.

**Language-Vision Pretraining Models.** Language-vision pretraining models, like CLIP, have shown promising results in open-world environments, making significant strides in the field of machine learning by creating a shared space for text and images (Radford et al., 2021; Li et al., 2022a; Singh et al., 2022; Lüddecke & Ecker, 2022; Frome et al., 2013; Socher et al., 2013; Elhoseiny et al., 2013). The advent of the Transformer architecture has propelled advancements in this area, paving the way for the development of more sophisticated models (Vaswani et al., 2017). However, these models exhibit a trade-off: while they excel in zero-shot generalization, they struggle with domain shifts and unseen categories during task-specific fine-tuning (Radford et al., 2021; Wortsman et al., 2022). Recent studies aim to optimize the fine-tuning process to mitigate this trade-off (Gao et al., 2021; Zhang et al., 2021a; Wortsman et al., 2022; Shu et al., 2022; Zhou et al., 2022b; Lu et al., 2022; Zhou et al., 2022a; Shu et al., 2023). Among the various fine-tuning methods, prompt learning stands out as an effective technique, but it is not without its challenges, such as the propensity for learned prompts to overfit to the training data, leading to suboptimal performance on unseen classes (Zhou et al., 2022a; Shu et al., 2023). Despite the progress made, persistent issues like attention misalignment and predictive category misalignment still hinder the stability of visual representations.

## 3 METHODOLOGY

In this section, we outline our approach in three distinct phases: Initially, we discuss the adaptation of CLIP models to handle open-world environment challenges (Section 3.1). Subsequently, we introduce Attention Alignment Loss, a novel objective function aimed at mitigating the attention misalignment of multi-head attention layers (Section 3.2). Finally, we present Semantic Label Smoothing, a new regularization designed to correct predictive category misalignment (Section 3.3).

### 3.1 ADAPTING CLIP IN OPEN-WORLD ENVIRONMENTS

**Problem Setup.** We start with a pre-trained CLIP model and adapt it using a task-specific training dataset $\mathcal{S} = \{(\mathbf{x}, y)\}$, where $y \in \mathcal{Y}$ represents the class label. Our primary aim is to fine-tune the CLIP model such that it performs robustly on unknown test data $\mathcal{T} = \{(\mathbf{x}', y')\}$. We focus on two distinct open-world challenges: *domain shifts*, characterized by a divergence in the data distribution $P(\mathbf{x}, y) \neq P(\mathbf{x}', y')$, and *unseen classes*, indicated by a difference in class labels $\mathcal{Y} \neq \mathcal{Y}'$. Rather than isolating these challenges, we venture into a more intricate *in-the-wild* setting where both open-world challenges may co-occur. Our approach is designed to be universally applicable, without making assumptions about specific types of distributional shifts and unknown categories, such as diversity or spurious shifts (Ye et al., 2022). Moreover, the condition $\mathcal{Y} \neq \mathcal{Y}'$ implies that the test data may introduce classes that were not present in the training set.

**CLIP Overview.** Our focus is on augmenting the open-world adaptability of the language-vision model CLIP (Radford et al., 2021) as it serves as a landmark contribution in the field. Unlike traditional models that rely on human-annotated labels, CLIP learns from raw text descriptions associated with images. The CLIP model is trained using a contrastive learning approach, which aims to bring similar image-text pairs closer in the embedding space while pushing dissimilar pairs

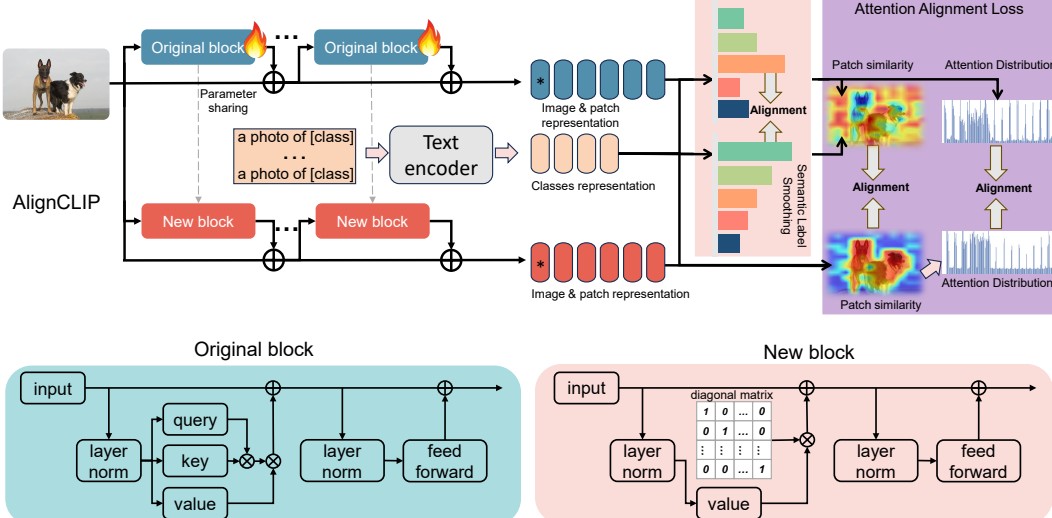

Figure 2: The AlignCLIP framework possesses a Siamese architecture. The first structure harmoniously integrates original attention blocks, while the second one is assembled with $\mathbf{D}$-V structure blocks. These new blocks employ a diagonal matrix, substituting the softmax operation as represented in equation 5. The new structure excels in producing coherent similarity distributions to help align the attention and similarity distributions of the first structure. Moreover, AlignCLIP calculates the class similarity matrix by leveraging the class text embedding from the text encoder, aiming to align the prediction distribution of the first structure meticulously.

apart. Specifically, a positive pair consists of an image and its corresponding text description, while negative pairs are formed by mismatching images and texts. CLIP consists of two main components: an image encoder $g_I(\cdot)$ and a text encoder $g_T(\cdot)$. The image encoder and the text encoder are usually Transformer-based models. One of the unique features of CLIP is its ability to perform zero-shot classification. CLIP employs a unique and efficient approach for classification tasks by utilizing its pre-trained text encoder to convert the names of the target dataset's classes into embedding vectors, thereby synthesizing a zero-shot classifier. During the inference phase, the image encoder of CLIP extracts features from the input image, which is then compared with the class embeddings generated by the text encoder to ascertain the most probable class to which the image belongs.

**Image Encoder Fine-Tuning.** Following Kumar et al. (2022); Shu et al. (2023), we adopt a text-prompt-driven strategy to enhance CLIP's adaptability. This strategy generates text prompts for each class, represented as $\mathbf{t}_c$. An example prompt might be "a photo of a [CLASS]", where [CLASS] is substituted by the class name. These prompts are then embedded using the text encoder to produce $\mathbf{T}_c = g_T(\mathbf{t}_c)$. For predictions, the image embedding $\mathbf{I_x} = g_I(\mathbf{x})$ is juxtaposed with class text embeddings. The probability of a sample $\mathbf{x}$ being in class $y$ is then:

$$P(y|\mathbf{x}) = \frac{\exp\left(S\left(\mathbf{I_x}, \mathbf{T}_y\right)/\tau\right)}{\sum_{c=1}^{C} \exp\left(S\left(\mathbf{I_x}, \mathbf{T}_c\right)/\tau\right)}, \qquad (1)$$

where $S(\cdot, \cdot)$ denotes the similarity metric between embeddings, and $\tau$ is a temperature factor. We employ the cosine similarity as per CLIP's training protocol. This fine-tuning approach leverages both image and text modalities to minimize cross-entropy losses. However, while image patterns remain diverse, the text corpus in downstream tasks is more limited. Fine-tuning the text encoder might compromise the pre-trained semantic relationships, leading to overfitting and reduced performance in unseen class scenarios. Following Shu et al. (2023), we fine-tune only the image encoder, keeping the text encoder static, and introduce our novel Attention Alignment Loss to improve CLIP's attention on objects of interest to further enhance the stability of representations in open-world environments.

### 3.2 ATTENTION ALIGNMENT LOSS

When directly applying misaligned attention to the class token $A_c(\mathbf{x})$ for predictions, the model is biased towards giving higher attention to the background of each instance $\mathbf{x}$ over the foreground.

This attention misalignment compromises the model's ability to generalize effectively in open-world scenarios. Conversely, the pre-trained text modality is rich in semantic information, which can be leveraged to quantify the semantic similarity between the text embedding of the corresponding class $y$ and the visual tokens. To address these issues, we propose utilizing this semantic similarity distribution to realign the attention distribution across each multi-head attention layer. For each training example $(\mathbf{x}, y)$, we primarily propose the Attention Alignment Loss (AAL) as

$$\mathcal{L}_{\text{AAL}} = \sum_i A_c(\mathbf{x}_i) \log \left( \frac{A_c(\mathbf{x}_i)}{G(\mathbf{x}_i)} \right), \tag{2}$$

where $A_c(\mathbf{x}_i)$ represents the averaged attention value of the $i$-th visual token of $\mathbf{x}$ that sum up from $L$ multiple multi-head attention layers. Specifically, $A_c(\mathbf{x}_i)$ is computed by

$$A_c(\mathbf{x}_i) = \frac{1}{L \times J} \sum_{l=1}^{L} \sum_{j=1}^{J} \frac{\exp \left( \frac{Q_{c,j}^l K_{i,j}^l}{\sqrt{d}} \right)}{\sum_{m=1}^{M} \exp \left( \frac{Q_{c,j}^l K_{m,j}^l}{\sqrt{d}} \right)}, \tag{3}$$

where we denote by $J$ the head number, $Q_{c,j}^l$ the query vector of the class token $c$ in the $j$-head attention of the $l$-th multi-head attention layer, and $K_{i,j}^l$ the key vector of the $i$-th visual token in the $j$-head attention of the $l$-th multi-head attention layer. Moreover, $M$ denotes the visual token number, usually $M = 16 \times 16$ except the class token, and $d$ is a scaling factor to improve the model stability. Further, $G(\mathbf{x})$ in equation 2 is the normalized cosine similarity distribution between $\mathbf{x}$ and the text embedding of $y$, and $G(\mathbf{x}_i)$ is the similarity between a visual token $\mathbf{I}_{\mathbf{x}}^i$ and text embedding $\mathbf{T}_y$ by

$$G(\mathbf{x}_i) = \frac{\exp \left( S \left( \mathbf{I}_{\mathbf{x}}^i, \mathbf{T}_y \right) / \tau \right)}{\sum_{m=1}^{M} \exp \left( S \left( \mathbf{I}_{\mathbf{x}}^m, \mathbf{T}_y \right) / \tau \right)}. \tag{4}$$

When the visual token embedding $\mathbf{I}_{\mathbf{x}}^i$ can present the actual semantics, $G(\mathbf{x})$ can be viewed as a guide attention distribution that can align the attention distribution of the class tokens of the multi-head attention layers. However, since the attention matrices of each attention layer are misalignment, they would mislead the value vectors of visual tokens by

$$\text{Attention}(Q, K, V) = \text{Softmax} \left( \frac{QK^T}{\sqrt{d_k}} \right) V, \tag{5}$$

where $d_k$ has the same dimension as the key vectors, which is also the same for the query vectors. Accordingly, the similarity distribution of visual token embedding $\mathbf{I}_{\mathbf{x}}^i$ is problematic, as shown in Figure 1(b). To solve this problem, CLIP Surgery (Li et al., 2023) is proposed to modify the Q-K-V structure into a V-V-V structure in which the $Q$ and $K$ vectors are replaced by the $V$ vectors. After modifying the attention layers, the V-V-V structure can output a reasonable similarity distribution. In this paper, we propose to modify the Q-K-V attention structure by the diagonal matrix-based D-V structure, which emphasizes the locality of $V$ vectors and is more efficient as it reduces computational complexity. Specifically, D-V structure refers to an innovative attention structure by replacing the softmax-normalized product of $Q$ and $K^T$ with a diagonal matrix. Furthermore, the D-V structure is capable of computing a rational similarity distribution $G_{\text{D-V}}(\mathbf{x}_i)$, which denotes the normalized attention distribution obtained by calculating the cosine similarity distance between visual token features, using the D-V structure, and transposed text features.

Based on that, we further propose a Siamese structure, as shown in Figure 2. The first branch is composed of original attention blocks, and the second branch is made up of D-V structure blocks. The second branch does not receive gradients and shares the parameters with the original blocks. Based on $G_{\text{D-V}}(\mathbf{x}_i)$, we align $G(\mathbf{x}_i)$ with it to refine the attention distribution further. Consequently, we newly propose the Attention Alignment Loss as

$$\mathcal{L}_{\text{AAL}} = \sum_i \left[ A_c(\mathbf{x}_i) \log \left( \frac{A_c(\mathbf{x}_i)}{G_{\text{D-V}}(\mathbf{x}_i)} \right) + G(\mathbf{x}_i) \log \left( \frac{G(\mathbf{x}_i)}{G_{\text{D-V}}(\mathbf{x}_i)} \right) \right]. \tag{6}$$

### 3.3 Semantic Label Smoothing

When using cross-entropy loss based on equation 1, the goal is to make image embeddings $\mathbf{I}_{\mathbf{x}}$ similar to the corresponding text embeddings $\mathbf{T}_y$ of their correct classes. However, this approach treats

all incorrect classes equally, ignoring any complex semantic relationships among them. This can lead to overconfidence and misclassification of images into semantically unrelated categories when errors occur. On the other hand, the pre-trained text modality has a rich semantic understanding of the relationships between different classes. Thus, we propose Semantic Label Smoothing (SLS), a specialized regularization technique that aligns the model's predicted category distribution with the semantic relationships among classes identified during the language-vision pre-training phase. One may argue the difference between SLS and TeS (Wang et al., 2023). TeS is a complex instance-level label smoothing technique using a projection head for diverse reference distributions, whereas SLS is a simpler class-level method focusing on representation stability in open-world environments.

Given the text embeddings of each class, we can construct a $C \times C$ class similarity matrix $\mathbf{S}$ by calculating the cosine similarity between class text embeddings, which serves as a quantifiable representation of the semantic relationships between different classes. One of the key advantages of utilizing this matrix is that it captures the inter-class relationships, thereby enabling the model to make more coherent predictions. Based on the similarity matrix, we formulate Semantic Label Smoothing as an extension of the standard label smoothing technique. The standard label smoothing equation is given by

$$\mathcal{L}_{\text{LS}} = (1 - \epsilon) \cdot \mathcal{L}_{\text{CE}} + \epsilon \cdot \mathcal{L}_{\text{U}}, \qquad (7)$$

where $\mathcal{L}_{\text{CE}}$ is the cross-entropy loss, $\mathcal{L}_{\text{U}}$ is the uniform distribution over classes, and $\epsilon$ is the smoothing factor. In SLS, we replace the uniform distribution $\mathcal{L}_{\text{U}}$ with a semantic distribution $\mathcal{L}_{\text{S}}$ derived from the similarity matrix. Thus, the SLS loss becomes

---

**Algorithm 1** Training Procedure of AlignCLIP

**Input:** Pre-trained CLIP image encoder $\theta_0$, learning rate $\eta$
  Initialize the Siamese structure and BMA model $\theta_0^{\text{BMA}} \leftarrow \theta_0$
  Calculate class similarity with text encoder
  **for** $t \in [1, T]$ **do**
    Sample data $\{(\mathbf{x}, y)\}$ from the training set $\mathcal{S}$
    Calculate AAL loss $\mathcal{L}_{\text{AAL}}$ as in equation 6
    Calculate SSL loss $\mathcal{L}_{\text{SLS}}$ as in equation 8
    Update model parameters $\theta_t \leftarrow \theta_{t-1} - \eta \nabla_{\theta_{t-1}} \mathcal{L}$
    Update the BMA model $\theta_t^{\text{BMA}}$ by $\theta_t$
  **end for**
**Output:** The final BMA model $\theta_T^{\text{BMA}}$

---

$$\mathcal{L}_{\text{SLS}} = (1 - \epsilon) \cdot \mathcal{L}_{\text{CE}} + \epsilon \cdot \mathcal{L}_{\text{S}}, \qquad (8)$$

where $\mathcal{L}_{\text{S}}$ is computed as

$$\mathcal{L}_{\text{S}} = \sum_i \mathbf{S}_{ij} \log(p(y_i|\mathbf{x})), \qquad (9)$$

where $\mathbf{S}_{ij}$ is the entry in the similarity matrix for class $i$ and class $j$, and $p(y_i|\mathbf{x})$ is the predicted probability of class $i$ given input $\mathbf{x}$. Details of the similarity matrix are available in the appendix B.5.

The full architecture of the AlignCLIP model is depicted in Figure 2, while the comprehensive training algorithm is outlined in Algorithm 1. The ultimate loss function for optimization is formulated as $\mathcal{L} = \mathcal{L}_{\text{AAL}} + \lambda \mathcal{L}_{\text{SLS}}$. During the fine-tuning process, both Attention Alignment Loss and Semantic Label Smoothing are applied at each optimization step to bolster the model's capabilities. To harmonize the trade-off between general applicability and task-specific performance, we incorporate a Beta Moving Average (BMA) strategy, as detailed in Shu et al. (2023). AAL and SLS are used only during the training phase. In the reference time, our method functions similarly to CLIP.

## 4 EXPERIMENTS

We explore three specialized categories of open-world experiments. The first category, referred to as *domain shift*, examines scenarios where the test dataset originates from a distribution distinct from that of the training set. The second category, termed *unseen class*, focuses on test samples that belong to classes absent from the training data. While the first two experiments are designed to examine each type of open-world challenge independently, the third experiment ventures into a more elaborate testing environment that unifies both the *domain shift* and *unseen class* challenges.

**Technical Details.** For our experiments, we employ the ViT-B/16 configuration of the pre-trained CLIP model as the image encoder (Dosovitskiy et al., 2021). The softmax temperature is retained at its pre-trained setting of $\tau = 0.01$. We use consistent hyper-parameters $\lambda = 1$ and $\epsilon = 0.1$ across all datasets. The $d$ and $d_k$ adhere to the ViT-B/16 default values. Training is conducted with a batch size

Table 1: Accuracy of models evaluated on the DomainBed benchmark in conditions of domain shift.

| Method | Backbone | PACS | VLCS | Office-Home | TerraInc | DomainNet | Avg. |
|---|---|---|---|---|---|---|---|
| ERM | ResNet | 85.5 | 77.5 | 66.5 | 46.1 | 40.9 | 63.3 |
| CORAL (2016) | ResNet | 86.2 | 78.8 | 68.7 | 47.6 | 41.5 | 64.6 |
| Zero-shot | CLIP | 96.2 | 81.7 | 82.0 | 32.7 | 57.5 | 70.2 |
| ERM | CLIP | $96.1_{\pm 0.5}$ | $83.0_{\pm 0.2}$ | $83.3_{\pm 0.3}$ | $\mathbf{60.9}_{\pm 0.2}$ | $59.9_{\pm 0.1}$ | $76.7_{\pm 0.2}$ |
| MIRO (2022b) | CLIP | 95.6 | 82.2 | 82.5 | 54.3 | 54.0 | 73.7 |
| DPL (2022) | CLIP | **97.3** | 84.3 | 84.2 | 52.6 | 56.7 | 75.0 |
| CLIPood (2023) | CLIP | $97.2_{\pm 0.1}$ | $84.1_{\pm 0.1}$ | $86.1_{\pm 0.1}$ | $59.4_{\pm 0.6}$ | $62.7_{\pm 0.1}$ | $77.9_{\pm 0.1}$ |
| AlignCLIP | CLIP | $\mathbf{97.3}_{\pm 0.2}$ | $\mathbf{85.1}_{\pm 0.2}$ | $\mathbf{86.9}_{\pm 0.1}$ | $59.5_{\pm 0.4}$ | $\mathbf{63.5}_{\pm 0.1}$ | $\mathbf{78.5}_{\pm 0.1}$ |

Table 2: Model accuracy on ImageNet under various domain shifts.

| Method | In-Distribution | Out-of-Distributions | | | | |
|---|---|---|---|---|---|---|
| | ImageNet | ImageNet-V2 | ImageNet-S | ImageNet-A | ImageNet-R | Avg. |
| Zero-shot | 66.7 | 60.8 | 46.1 | 47.8 | 74.0 | 57.2 |
| Fine-tune | $68.2_{\pm 0.1}$ | $61.9_{\pm 0.1}$ | $46.8_{\pm 0.1}$ | $46.4_{\pm 0.1}$ | $75.1_{\pm 0.1}$ | $57.6_{\pm 0.1}$ |
| CoOp (2022b) | 71.5 | 64.2 | 48.0 | 49.7 | 75.2 | 59.3 |
| CoCoOp (2022a) | 71.0 | 64.2 | 48.8 | $\mathbf{50.6}$ | 76.2 | 59.9 |
| CLIPood (2023) | $72.3_{\pm 0.1}$ | $64.5_{\pm 0.1}$ | $47.6_{\pm 0.1}$ | $46.8_{\pm 0.1}$ | $75.6_{\pm 0.1}$ | $58.6_{\pm 0.1}$ |
| AlignCLIP | $\mathbf{73.2}_{\pm 0.1}$ | $\mathbf{64.9}_{\pm 0.1}$ | $\mathbf{49.2}_{\pm 0.1}$ | $49.8_{\pm 0.1}$ | $\mathbf{77.3}_{\pm 0.1}$ | $\mathbf{60.3}_{\pm 0.1}$ |

of 36 and features random resized cropping. We utilize the AdamW optimizer (Loshchilov & Hutter, 2019) and adopt a cosine learning rate schedule. By default, the learning rate is set at $5 \times 10^{-6}$ and the training spans $5,000$ iterations. The reported AlignCLIP results include both the average and standard deviation from three separate runs with varying random seeds.

## 4.1 GENERALIZE CLIP TO DOMAIN SHIFT

**Benchmarks.** We evaluate open-world adaptability using two benchmarks. The first employs five datasets from DomainBed (Gulrajani & Lopez-Paz, 2021), including PACS, VLCS, Office-Home, TerraIncognita, and DomainNet. We follow DomainBed's train-validate-test split and use a leave-one-out strategy for open-world testing. The second benchmark uses ImageNet (Deng et al., 2009) for training and tests on four of its variants with distribution shifts: ImageNet-V2, ImageNet-Sketch, ImageNet-A, and ImageNet-R (Hendrycks et al., 2021a). We adopt a 16-shot training subset for each variant, as per (Zhou et al., 2022a), while using the original test sets for evaluation.

**Results.** Table 1 shows average test accuracy on the DomainBed benchmark. We compare AlignCLIP with methods using CLIP and ResNet-50 pre-trained models, including zero-shot, ERM-based fine-tuning, and state-of-the-art techniques like MIRO, DPL, and CLIPood. AlignCLIP significantly outperforms ResNet-based methods and even bests existing CLIP-based state-of-the-art models, highlighting its effectiveness in domain shifts. In Table 2, AlignCLIP also excels in various ImageNet variants, outperforming leading methods such as CLIPood in open-world scenarios.

## 4.2 GENERALIZE CLIP TO UNSEEN CLASS

**Benchmarks.** For evaluating unseen class recognition, we use diverse benchmarks covering multiple tasks. These include general object classification with ImageNet and Caltech101, fine-grained tasks like OxfordPets, StanfordCars, Flowers102, Food101, and FGVCAircraft, and specialized tasks such as SUN397 for scenes, UCF101 for actions, DTD for textures, and EuroSAT for satellite images. Following the protocol in Zhou et al. (2022a), we split each dataset's classes into base and unseen categories. The model is trained on base classes and evaluated on both to assess representation stability. Specifically, 50% of classes are as base, while the remaining serve as unseen categories.

**Results.** Table 3 reports AlignCLIP's performance in open-class scenarios. We report accuracies for both base and unseen classes, along with their harmonic mean (H) (Fu et al., 2020) to highlight the model's balanced performance across both. AlignCLIP is compared with zero-shot CLIP, as well as state-of-the-art methods like CoOp, CoCoOp, and CLIPood. Detailed dataset-specific results are

Table 3: Evaluation of real-world performance on 11 downstream datasets featuring unseen classes.

(a) Average over 11 datasets

| | Base | New | H |
|---|---|---|---|
| CLIP | 69.3 | 74.2 | 71.7 |
| CoOp (2022b) | 82.7 | 63.2 | 71.7 |
| CoCoOp (2022a) | 80.5 | 71.7 | 75.8 |
| CLIPood | $83.7_{\pm0.1}$ | $74.2_{\pm0.1}$ | $78.4_{\pm0.1}$ |
| AlignCLIP | $\mathbf{84.5}_{\pm0.1}$ | $\mathbf{74.8}_{\pm0.1}$ | $\mathbf{79.0}_{\pm0.1}$ |

(b) ImageNet

| | Base | New | H |
|---|---|---|---|
| CLIP | 72.4 | 68.1 | 70.2 |
| CoOp (2022b) | 76.5 | 67.9 | 71.9 |
| CoCoOp (2022a) | 76.0 | 70.4 | 73.1 |
| CLIPood | $77.5_{\pm0.1}$ | $69.9_{\pm0.1}$ | $73.5_{\pm0.1}$ |
| AlignCLIP | $\mathbf{77.8}_{\pm0.1}$ | $\mathbf{70.6}_{\pm0.1}$ | $\mathbf{74.1}_{\pm0.1}$ |

Table 4: Accuracy on Office-Home and DomainNet with both domain shift and unseen classes.

| Method | Office-Home | | | | DomainNet | | | | | |
|---|---|---|---|---|---|---|---|---|---|---|
| | A | C | P | R | C | I | P | Q | R | S |
| CLIP | 82.6 | 67.3 | 88.8 | 89.5 | 71.4 | 47.1 | 66.2 | 13.8 | 83.4 | 63.4 |
| CoOp | $82.7_{\pm0.5}$ | $67.2_{\pm0.7}$ | $90.2_{\pm1.0}$ | $89.2_{\pm0.6}$ | $73.4_{\pm0.3}$ | $51.8_{\pm0.3}$ | $67.9_{\pm1.0}$ | $13.7_{\pm0.8}$ | $83.9_{\pm0.5}$ | $66.0_{\pm0.2}$ |
| CLIPood | $82.5_{\pm0.1}$ | $68.2_{\pm0.2}$ | $90.5_{\pm0.2}$ | $89.7_{\pm0.1}$ | $75.2_{\pm0.1}$ | $54.4_{\pm0.1}$ | $70.0_{\pm0.1}$ | $19.1_{\pm0.1}$ | $84.5_{\pm0.1}$ | $67.9_{\pm0.1}$ |
| AlignCLIP | $\mathbf{83.5}_{\pm0.1}$ | $\mathbf{70.1}_{\pm0.2}$ | $\mathbf{91.0}_{\pm0.1}$ | $\mathbf{91.2}_{\pm0.1}$ | $\mathbf{75.7}_{\pm0.1}$ | $\mathbf{55.7}_{\pm0.1}$ | $\mathbf{71.1}_{\pm0.1}$ | $\mathbf{19.2}_{\pm0.3}$ | $\mathbf{84.6}_{\pm0.1}$ | $\mathbf{70.0}_{\pm0.1}$ |

available in the appendix B.1. Our findings indicate that while CoOp suffers a notable decline in accuracy for unseen classes after adaptation, CoCoOp maintains better performance but compromises on base classes. Both methods fall short of zero-shot performance for unseen classes. CLIPood improves this by fully fine-tuning the image encoder, validating the effectiveness of comprehensive parameter tuning. In specific datasets like ImageNet, AlignCLIP, along with CoCoOp and CLIPood, improves unseen class performance by adapting the model using related base classes. Overall, AlignCLIP outperforms existing methods and zero-shot predictions, showcasing its ability to excel in downstream tasks while preserving open-class generalization.

## 4.3 GENERALIZE CLIP TO DOMAIN SHIFT AND UNSEEN CLASS

**Benchmarks.** We use Office-Home and DomainNet from DomainBed to simulate open-world environments involving both domain shifts and unseen classes. Classes are split into training base and testing unseen subsets. Using a leave-one-domain-out approach, we train on base classes and test on a mix of base and unseen classes, reflecting a typical open-world setting.

**Results.** Table 4 shows domain-specific accuracy. AlignCLIP is benchmarked against zero-shot CLIP, CoOp, and CLIPood. CoOp shows mixed results, excelling in some domains but falling short in others. CLIPood further improves the performance by fine-tuning the image encoder. AlignCLIP consistently outperforms all benchmarks, confirming its robustness in open-world environments.

## 4.4 ANALYSIS OF ALIGNCLIP

**Ablation Study.** We examine the impact of Attention Alignment Loss (AAL) and Semantic Label Smoothing (SLS) in AlignCLIP using the Office-Home and VLCS datasets. We test AlignCLIP variants with and without these modules in two open-world settings: domain shift alone (Domain) and a combination of domain shift and unseen classes (Domain+Class). Figure 3a shows that incorporating AAL and SLS significantly boosts performance in both scenarios. The results validate the modules' effectiveness in enhancing open-world generalization. Optimal performance is achieved when both AAL and SLS are included, indicating their synergistic role in robust open-world generalization. While AAL focuses on optimizing the attention mechanism to better focus on relevant features, SLS aims to refine the prediction mechanism to be more semantically accurate and less prone to misclassification. Together, they address different aspects of the model's performance but work towards the same goal of improving stability and reliability in complex, real-world scenarios.

**Analysis on Attention Alignment Loss.** We delve into the effectiveness of AAL by examining its influence on the alignment of multi-head attention layer distributions (AL) and the alignment of similarity distribution (AS). Figure 3b reveals that aligning both AL and AS yields optimal performance. Further, Figure 3c indicates that aligning just the first four layers suffices for best results because the attention distributions in the deeper layers are very sparser. Language-vision models often have an attention bias towards background visual tokens, as shown in the lower row of Figure 4.

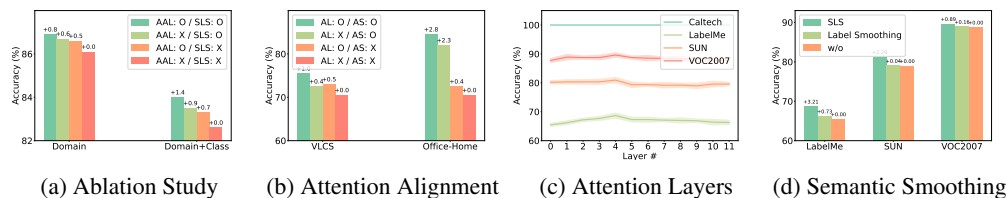

(a) Ablation Study     (b) Attention Alignment     (c) Attention Layers     (d) Semantic Smoothing

Figure 3: Analysis experiments on AlignCLIP. AL and AS in (b) represent the alignment of layer attention and the alignment of similarity distribution, respectively.

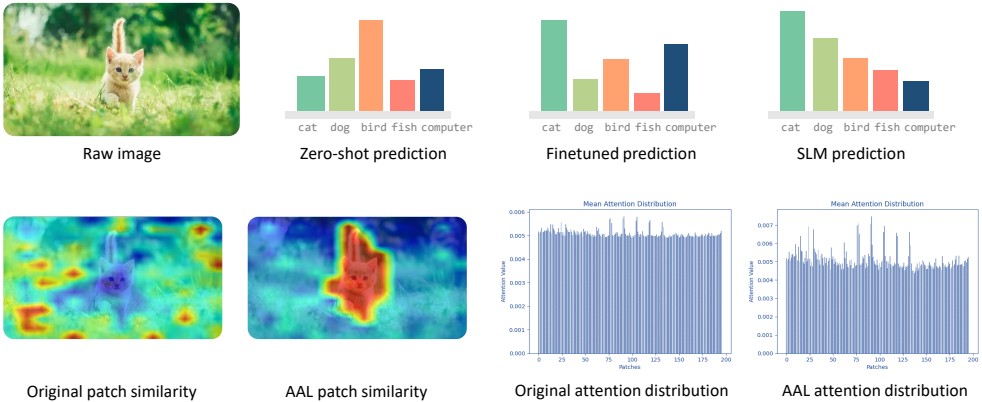

Figure 4: Case study of AlignCLIP. The first row shows the predictions from models of zero-shot and fine-tuned with and without semantic label smoothing. The second row shows the attention distribution and token similarity from models trained with attention alignment loss.

This bias hampers robustness, especially when encountering new distributions or categories. AAL effectively mitigates this issue, balancing attention between base and unseen classes. Thus, AAL stands as a key innovation in AlignCLIP, significantly enhancing its out-of-distribution performance.

**Analysis on Semantic Label Smoothing.** We scrutinize the effectiveness of SLS against conventional Label Smoothing techniques in Figure 3d. SLS shows improvements of 3.21%, 2.29%, and 0.89% in the LabelMe, SUN, and VOC2007 datasets. To visualize the impact of SLS, we showcase the top-five class predictions on test images in Figure 4 for models trained with and without SLS. For instance, when faced with an image of an untrained `cat` class, the model without SLS assigns higher output probabilities to unrelated categories like `computer` over semantically closer ones like `dog`. This indicates a semantic misalignment. Conversely, the model with SLS not only generates semantically coherent predictions but also accurately classifies the `cat`. This underscores SLS's capability to maintain semantic relationships among classes, thereby enhancing the model's generalization across both domain shifts and unseen classes. Consequently, SLS proves to be a pivotal element in the AlignCLIP framework, substantially elevating its performance in open-world scenarios.

## 5 CONCLUSION

In this paper, we address the pressing need for a unified approach to tackle both domain shift and unseen classes in open-world environments, particularly in the context of downstream tasks. We introduce AlignCLIP, a fine-tuning methodology aimed at enhancing the representation stability of CLIP models. AlignCLIP employs a novel training objective, Attention Alignment Loss, to synchronize the attention distributions across multi-head attention layers and to align the correlation between visual tokens and text descriptors. Additionally, we introduce Semantic Label Smoothing, a technique designed to maintain a prediction hierarchy based on class similarity, leveraging the rich textual information available in the pre-trained text modality. Our empirical evaluations across a diverse range of datasets and open-world conditions conclusively demonstrate that AlignCLIP consistently outperforms existing state-of-the-art generalization approaches. Through AlignCLIP, we offer a robust and effective solution for enhancing open-world generalization, thereby contributing to the broader applicability and reliability of CLIP models in open-world scenarios.

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

# A Experimental Details

## A.1 Technical Details

Default Experiment Configurations:

- **Model Configuration:** ViT-B/16 variant of the pre-trained CLIP model as the image encoder (Dosovitskiy et al., 2021).

- **Softmax Temperature:** Maintained at its pre-trained value of $\tau = 0.01$.

- **Hyper-parameters:** Consistently applied across all datasets with $\lambda = 1$ and $\epsilon = 0.1$.

- **Parameters:** $d$ and $d_k$ set to the default values of the ViT-B/16 configuration.

- **Aligned Layers:** 4.

- **Batch Size:** 36.

- **Image Augmentation:** Incorporation of random resized cropping.

- **Optimizer:** AdamW algorithm (Loshchilov & Hutter, 2019).

- **Learning Rate Schedule:** Cosine.

- **Standard Learning Rate:** $5 \times 10^{-6}$.

- **Standard Training Duration:** 5,000 iterations.

- **Results Reporting:** Mean and standard deviation derived from three independent runs with different random seeds.

**Note:** While the aforementioned configurations act as our default starting point for all experiments, specific datasets might necessitate adjustments in terms of the number of iterations and learning rates, as detailed in Shu et al. (2023).

Implementation Details:

- **Framework:** PyTorch 2.0

- **Library:** torchvision 0.15.1

- **Compute:** CUDA 11.8

- **Hardware:** NVIDIA A100 GPUs on a cluster.

## A.2 Prompt Templates for Each Dataset

We adopt a default prompt template for class labels: "a photo of [CLASS].", where [CLASS] is replaced by the class name, with hyphens substituted by spaces. For fine-grained classification datasets, as suggested by Zhou et al. (2022b); Shu et al. (2023), we incorporate the superclass name or a descriptive phrase into the template. The dataset-specific templates are:

- **OxfordPets:** "a photo of a [CLASS], a type of pet."

- **FGVCAircraft:** "a photo of a [CLASS], a type of aircraft."

- **DTD:** "[CLASS] texture."

- **EuroSAT:** "a centered satellite photo of [CLASS]."

- **Food101:** "a photo of a [CLASS], a type of food."

- **UCF101:** "a photo of a person doing [CLASS]."

- **Other datasets:** "a photo of a [CLASS]."

A.2.1 DATASET DETAILS

**Domain Shift Datasets:** Our assessment of open-world generalization in the context of domain shifts is conducted using two key benchmarks. The first benchmark incorporates five multi-domain datasets from DomainBed (Gulrajani & Lopez-Paz, 2021), including PACS (Li et al., 2017), VLCS (Torralba & Efros, 2011), Office-Home (Venkateswara et al., 2017), TerraIncognita (Beery et al., 2018), and DomainNet (Peng et al., 2019). For each dataset, we adhere to the train-validate-test partitioning scheme specified by the DomainBed framework. We employ a leave-one-out evaluation strategy, wherein a single domain is isolated for testing open-world generalization capabilities, while the remaining domains serve as the training set. For our second benchmark, we utilize ImageNet (Deng et al., 2009) as the foundational training dataset. We then assess model performance across four modified versions of ImageNet, each introducing different types of distribution shifts: ImageNet-V2 (Recht et al., 2019), ImageNet-Sketch (Wang et al., 2019), ImageNet-A (Hendrycks et al., 2021b), and ImageNet-R (Hendrycks et al., 2021a). In line with the methodology outlined in Zhou et al. (2022a), we randomly select a 16-shot training subset from each variant while retaining the original test sets for performance evaluation. The details of these datasets are as follows.

- **PACS (Li et al., 2017):** A dataset designed for domain generalization, PACS comprises 10,091 images across four domains—Photo, Art Painting, Cartoon, and Sketch, each with seven categories.

- **VLCS (Torralba & Efros, 2011):** VLCS is a multifaceted dataset aimed at domain generalization, integrating five shared categories from four diverse datasets: Caltech, LabelMe, SUN, and VOC2007.

- **Office-Home (Venkateswara et al., 2017):** Created to assess domain adaptation algorithms, Office-Home encompasses four domains—Artistic, Product, Real-World Images, and Clip Art, each hosting 64 categories.

- **DomainNet (Peng et al., 2019):** DomainNet is a comprehensive dataset for domain generalization, featuring six domains and encompassing 345 categories of common objects.

- **TerraInc (Beery et al., 2018):** TerraInc consists of images from twenty camera traps, focused on learning recognition in specific locations and generalizing to new ones to monitor animal populations.

- **ImageNet-V2 (Recht et al., 2019):** A test set extension of ImageNet, ImageNet-V2 consists of images collected following the original labeling protocol, with 10 images per class.

- **ImageNet-S (Wang et al., 2019):** ImageNet-S contains 1,183,322 training, 12,419 validation, and 27,423 testing images, distributed across 919 categories.

- **ImageNet-A (Hendrycks et al., 2021b):** ImageNet-A is comprised of images labeled with ImageNet labels, selectively including those images that ResNet-50 models incorrectly classify.

- **ImageNet-R (Hendrycks et al., 2021a):** ImageNet-R incorporates various renditions of 200 ImageNet classes, including art, cartoons, and sculptures, totaling 30,000 images.

**Unseen Category Datasets:** To assess the model's ability to generalize to unseen classes, we employ a comprehensive benchmark that spans a wide array of recognition tasks. These include general object classification tasks, represented by ImageNet (Deng et al., 2009) and Caltech101 (Fei-Fei et al., 2004), as well as fine-grained classification tasks such as OxfordPets (Parkhi et al., 2012), StanfordCars (Krause et al., 2013), Flowers102 (Nilsback & Zisserman, 2008), Food101 (Bossard et al., 2014), and FGVCAircraft (Maji et al., 2013). We also include specialized classification tasks like SUN397 (Xiao et al., 2010) for scene recognition, UCF101 (Soomro et al., 2012) for action recognition, DTD (Cimpoi et al., 2014) for texture classification, and EuroSAT (Helber et al., 2019) for satellite image categorization. In line with the protocol established in Zhou et al. (2022a), we divide the classes in each dataset into two subsets: base classes and unseen classes. The model is trained on data from the base classes and subsequently tested on both base and unseen classes to gauge its generalization capabilities. The details of these datasets are as follows.

- **ImageNet (Deng et al., 2009):** ImageNet is a comprehensive dataset featuring 14,197,122 annotated images, categorized into 1,000 classes according to the WordNet hierarchy.

- **Caltech101 (Fei-Fei et al., 2004):** Caltech101 offers around 9k variable-sized images across 101 object categories and one background clutter class, with each image labeled with a single object.
- **OxfordPets (Parkhi et al., 2012):** Developed by the Visual Geometry Group at Oxford, OxfordPets is a pet dataset consisting of approximately 200 images in each of its 37 categories.
- **StanfordCars (Krause et al., 2013):** StanfordCars presents 16,185 images distributed across 196 car classes, with a nearly equal split between training and testing images.
- **Flowers102 (Nilsback & Zisserman, 2008):** Flowers102 features images of flowers commonly found in the United Kingdom, distributed across 102 categories, each containing between 40 and 258 images.
- **Food101 (Bossard et al., 2014):** Food101 is a culinary dataset encompassing 101,000 images across 101 food categories, with each class having 250 test images and 750 training images.
- **FGVCAircraft (Maji et al., 2013):** FGVCAircraft is an aviation-centric dataset, comprising 10,200 images spanning 102 different aircraft model variants.
- **SUN397 (Xiao et al., 2010):** SUN397 is a diverse scene understanding dataset, containing 108,753 images spread over 397 categories, designed for the Scene UNderstanding (SUN) benchmark.
- **DTD (Cimpoi et al., 2014):** DTD is a texture-centric database, organized into 47 categories inspired by human perception, and consists of 5640 images.
- **EuroSAT (Helber et al., 2019):** EuroSAT, based on Sentinel-2 satellite images, includes 27,000 labeled and geo-referenced samples across 10 classes, covering 13 spectral bands.
- **UCF101 (Soomro et al., 2012):** UCF101 is a realistic action recognition dataset, comprising videos from YouTube across 101 action categories.

**Domain Shift and Unseen Category Datasets:** To simulate a more authentic, real-world scenario that combines both domain shift and unseen classes, we select Office-Home and DomainNet from the DomainBed suite, owing to their ample class diversity suitable for unseen class evaluation. We partition the classes in each dataset into two subsets: base classes for training and unseen classes for testing. Following a leave-one-domain-out strategy, we train the model on base-class data from the training domains and evaluate it on a test set that includes both base and unseen classes. This approach mimics a more realistic open-world testing environment, as it is often impractical to pre-determine whether test data will belong to base or unseen classes.

**License Information:** OxfordPets is under the CC BY-SA 4.0 license. Other datasets are publicly available online and under custom licenses for non-commercial usage.

## B MORE EXPERIMENTAL RESULTS

### B.1 DETAILED RESULTS ON UNSEEN CLASSES

This section delineates comprehensive results related to the generalization to unseen classes, as elaborated in Section 4.2. The results for each individual dataset, along with the cumulative average across all 11 datasets, are meticulously cataloged in Table 5.

AlignCLIP exhibits exemplary adaptation performance on base classes across the majority of the datasets, effectively diminishing the disparity in performance observed in zero-shot predictions on unseen classes, and often exceeding them. This superior performance is attributed to the innovative Attention Alignment Loss and Semantic Label Smoothing techniques incorporated in AlignCLIP, emphasizing its enhanced unseen class generalization ability.

When the results are averaged, it is evident that AlignCLIP markedly surpasses both zero-shot predictions and other prevailing methodologies. This accentuates AlignCLIP's dual proficiency in optimizing model adaptation to augment performance on downstream tasks while maintaining robust open-world generalization capability on unseen classes.

Table 5: Performance on generalization is evaluated across 11 downstream datasets, each incorporating unseen classes.

(a) Average over 11 datasets

|  | BASE | NEW | H |
|---|---|---|---|
| CLIP | 69.3 | 74.2 | 71.7 |
| CoOp | 82.7 | 63.2 | 71.7 |
| CoCoOp | 80.5 | 71.7 | 75.8 |
| CLIPood | $83.7_{\pm0.1}$ | $74.2_{\pm0.1}$ | $78.4_{\pm0.1}$ |
| AlignCLIP | $\mathbf{84.5}_{\pm0.1}$ | $\mathbf{74.8}_{\pm0.1}$ | $\mathbf{79.0}_{\pm0.1}$ |

(b) ImageNet

|  | BASE | NEW | H |
|---|---|---|---|
| CLIP | 72.4 | 68.1 | 70.2 |
| CoOp | 76.5 | 67.9 | 71.9 |
| CoCoOp | 76.0 | 70.4 | 73.1 |
| CLIPood | $77.5_{\pm0.1}$ | $69.9_{\pm0.1}$ | $73.5_{\pm0.1}$ |
| AlignCLIP | $\mathbf{77.8}_{\pm0.1}$ | $\mathbf{70.6}_{\pm0.1}$ | $\mathbf{74.1}_{\pm0.1}$ |

(c) Caltech101

|  | BASE | NEW | H |
|---|---|---|---|
| CLIP | 96.8 | 94.0 | 95.4 |
| CoOp | 98.0 | 89.8 | 93.7 |
| CoCoOp | 98.0 | 93.8 | 95.8 |
| CLIPood | $98.5_{\pm0.1}$ | $93.9_{\pm0.4}$ | $96.2_{\pm0.2}$ |
| AlignCLIP | $\mathbf{98.5}_{\pm0.3}$ | $\mathbf{94.5}_{\pm0.3}$ | $\mathbf{96.5}_{\pm0.2}$ |

(d) OxfordPets

|  | BASE | NEW | H |
|---|---|---|---|
| CLIP | 91.2 | 97.3 | 94.1 |
| CoOp | 93.7 | 95.3 | 94.5 |
| CoCoOp | 95.2 | $\mathbf{97.7}$ | 96.4 |
| CLIPood | $95.5_{\pm0.2}$ | $97.1_{\pm0.2}$ | $96.3_{\pm0.1}$ |
| AlignCLIP | $\mathbf{96.4}_{\pm0.1}$ | $96.7_{\pm0.1}$ | $\mathbf{96.5}_{\pm0.1}$ |

(e) StanfordCars

|  | BASE | NEW | H |
|---|---|---|---|
| CLIP | 63.4 | $\mathbf{74.9}$ | 68.7 |
| CoOp | 78.1 | 60.4 | 68.1 |
| CoCoOp | 70.5 | 73.6 | 72.0 |
| CLIPood | $\mathbf{78.5}_{\pm0.1}$ | $72.4_{\pm0.2}$ | $75.4_{\pm0.2}$ |
| AlignCLIP | $78.4_{\pm0.3}$ | $73.9_{\pm0.2}$ | $\mathbf{76.1}_{\pm0.2}$ |

(f) Flowers102

|  | BASE | NEW | H |
|---|---|---|---|
| CLIP | 72.1 | $\mathbf{77.8}$ | 74.8 |
| CoOp | $\mathbf{97.6}$ | 59.7 | 74.1 |
| CoCoOp | 94.9 | 71.8 | 81.7 |
| CLIPood | $94.8_{\pm0.1}$ | $72.6_{\pm0.7}$ | $82.2_{\pm0.5}$ |
| AlignCLIP | $95.0_{\pm0.2}$ | $73.9_{\pm0.2}$ | $\mathbf{83.1}_{\pm0.2}$ |

(g) Food101

|  | BASE | NEW | H |
|---|---|---|---|
| CLIP | 90.1 | 91.2 | 90.7 |
| CoOp | 88.3 | 82.3 | 85.2 |
| CoCoOp | 90.7 | 91.3 | 91.0 |
| CLIPood | $90.5_{\pm0.1}$ | $91.1_{\pm0.1}$ | $90.8_{\pm0.1}$ |
| AlignCLIP | $\mathbf{91.3}_{\pm0.1}$ | $\mathbf{91.9}_{\pm0.1}$ | $\mathbf{91.6}_{\pm0.1}$ |

(h) FGVCAircraft

|  | BASE | NEW | H |
|---|---|---|---|
| CLIP | 27.2 | 36.3 | 31.1 |
| CoOp | 40.4 | 22.3 | 28.8 |
| CoCoOp | 33.4 | 23.7 | 27.7 |
| CLIPood | $42.1_{\pm0.1}$ | $35.9_{\pm0.3}$ | $38.8_{\pm0.2}$ |
| AlignCLIP | $\mathbf{42.8}_{\pm0.2}$ | $\mathbf{36.8}_{\pm0.1}$ | $\mathbf{39.6}_{\pm0.2}$ |

(i) SUN397

|  | BASE | NEW | H |
|---|---|---|---|
| CLIP | 69.4 | 75.4 | 72.2 |
| CoOp | 80.6 | 65.9 | 72.5 |
| CoCoOp | 79.7 | 76.9 | 78.3 |
| CLIPood | $80.2_{\pm0.1}$ | $78.9_{\pm0.1}$ | $79.5_{\pm0.1}$ |
| AlignCLIP | $\mathbf{81.5}_{\pm0.1}$ | $\mathbf{79.4}_{\pm0.1}$ | $\mathbf{80.4}_{\pm0.1}$ |

(j) DTD

|  | BASE | NEW | H |
|---|---|---|---|
| CLIP | 53.2 | $\mathbf{59.9}$ | 56.4 |
| CoOp | 79.4 | 41.2 | 54.2 |
| CoCoOp | 77.0 | 56.0 | 64.9 |
| CLIPood | $80.5_{\pm0.9}$ | $59.6_{\pm0.2}$ | $\mathbf{68.5}_{\pm0.3}$ |
| AlignCLIP | $\mathbf{81.8}_{\pm0.4}$ | $58.5_{\pm0.8}$ | $68.2_{\pm0.1}$ |

(k) EuroSAT

|  | BASE | NEW | H |
|---|---|---|---|
| CLIP | 56.5 | 64.1 | 60.0 |
| CoOp | 92.2 | 54.7 | 68.7 |
| CoCoOp | 87.5 | 60.0 | 71.2 |
| CLIPood | $97.2_{\pm0.1}$ | $67.1_{\pm0.4}$ | $79.4_{\pm0.3}$ |
| AlignCLIP | $\mathbf{98.9}_{\pm0.1}$ | $\mathbf{67.6}_{\pm0.4}$ | $\mathbf{80.4}_{\pm0.5}$ |

(l) UCF101

|  | BASE | NEW | H |
|---|---|---|---|
| CLIP | 70.5 | 77.5 | 73.9 |
| CoOp | 84.7 | 56.1 | 67.5 |
| CoCoOp | 82.3 | 73.5 | 77.6 |
| CLIPood | $86.1_{\pm0.2}$ | $78.4_{\pm0.2}$ | $82.1_{\pm0.1}$ |
| AlignCLIP | $\mathbf{87.0}_{\pm0.4}$ | $\mathbf{79.4}_{\pm0.4}$ | $\mathbf{83.0}_{\pm0.4}$ |

## B.2 DETAILED RESULTS ON DISTRIBUTION SHIFTS

In Table 6, we meticulously outline the results obtained from VLCS, PACS, Office-Home, and TerraIncognita datasets, shedding light on the substantial enhancements that AlignCLIP imparts

Table 6: We present detailed results for VLCS, PACS, Office-Home, and TerraIncognita, emphasizing the enhancements AlignCLIP brings to the zero-shot capabilities of CLIP under domain shifts. Our findings indicate a notable improvement in performance, with each dataset showcasing significant advancements. The most improved result for each dataset is highlighted to underscore the efficacy of AlignCLIP in addressing domain shifts.

| VLCS | Caltech101 | LabelMe | SUN09 | VOC2007 | Avg |
|---|---|---|---|---|---|
| CLIP Zero-shot | 100 | 67.9 | 72.7 | 86.5 | 81.7 |
| AlignCLIP | + 0.0 | + 1.0 | + 8.2 | + 4.3 | + 3.4 |
| **PACS** | **Art** | **Cartoon** | **Photo** | **Sketch** | **Avg** |
| CLIP Zero-shot | 97.3 | 99.1 | 100 | 88.2 | 96.2 |
| AlignCLIP | + 1.6 | + 0.7 | + 0.0 | + 1.9 | + 1.1 |
| **Office-Home** | **Artistic** | **Clipart** | **Product** | **Real World** | **Avg** |
| CLIP Zero-shot | 83.5 | 66.8 | 88.2 | 90.4 | 82.0 |
| AlignCLIP | + 4.5 | + 6.2 | + 4.7 | + 3.9 | + 4.9 |
| **TerraInc** | **location 100** | **location 38** | **location 43** | **location 46** | **Avg** |
| CLIP Zero-shot | 50.8 | 24.0 | 30.4 | 25.9 | 33.4 |
| AlignCLIP | + 20.4 | + 34.0 | + 28.5 | + 24.0 | + 26.7 |

to the inherent zero-shot capabilities of the original CLIP model, especially in scenarios involving domain shifts. Our meticulous analysis and empirical findings across these diverse datasets reveal a remarkable improvement in the model's performance, with each dataset manifesting substantial advancements due to the incorporation of AlignCLIP. These advancements are not just numerical but are indicative of the model's enhanced ability to adapt and generalize across varying domains and under different conditions, showcasing the robustness and versatility of AlignCLIP. To emphasize the efficacy and impact of AlignCLIP in mitigating the challenges posed by domain shifts, we have highlighted the most improved result within each dataset. These highlighted improvements serve as a testament to AlignCLIP's capability to significantly enhance the model's performance and adaptability across different domains. Particularly in the TerraInc dataset, which represents a natural shift scenario, AlignCLIP exhibits a substantial improvement of 26.7% in accuracy. This improvement is not merely a statistical increment but a demonstration of AlignCLIP's practical applicability and its potential to be deployed in real-world scenarios where domain shifts are prevalent, thereby validating its practical utility and effectiveness in real-world applications.

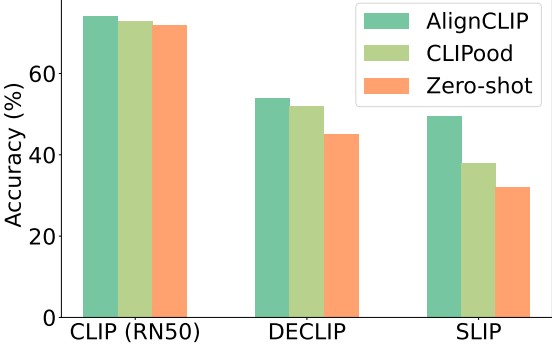

Figure 5: The results of AlignCLIP, CLIPood, and zero-shot approaches on other language-vision models.

Table 7: The results of Empirical Risk Minimization (ERM) with various backbone networks on the Domain Generalization (DG) benchmark are presented, with the data sourced from Zhang et al. (2022). The symbol $\dagger$ denotes that the figures are extracted from Table 2 in Iwasawa & Matsuo (2021), and * represents numbers sourced from MIRO (Cha et al., 2022a). The highest scores are highlighted in bold, and the runner-up scores are underlined for clarity and emphasis.

| Backbone Model | VLCS | PACS | Office-Home | Terra | Avg |
|---|---|---|---|---|---|
| ResNet18$^\dagger$ | $73.2 \pm 0.9$ | $80.3 \pm 0.4$ | $55.7 \pm 0.2$ | $40.7 \pm 0.3$ | 62.5 |
| ResNet50$^\dagger$ | $75.5 \pm 0.1$ | $83.9 \pm 0.2$ | $64.4 \pm 0.2$ | $45.4 \pm 1.2$ | 67.3 |
| Mixer-L16$^\dagger$ | $76.4 \pm 0.2$ | $81.3 \pm 1.0$ | $69.4 \pm 1.6$ | $37.1 \pm 0.4$ | 66.1 |
| BiT-M-R50x3$^\dagger$ | $76.7 \pm 0.1$ | $84.4 \pm 1.2$ | $69.2 \pm 0.6$ | $52.5 \pm 0.3$ | 70.7 |
| BiT-M-R101x3$^\dagger$ | $75.0 \pm 0.6$ | $84.0 \pm 0.7$ | $67.7 \pm 0.5$ | $47.8 \pm 0.8$ | 68.6 |
| BiT-M-R152x2$^\dagger$ | $76.7 \pm 0.3$ | $85.2 \pm 0.1$ | $71.3 \pm 0.6$ | $51.4 \pm 0.6$ | 71.1 |
| ViT-B16$^\dagger$ | $79.2 \pm 0.3$ | $85.7 \pm 0.1$ | $78.4 \pm 0.3$ | $41.8 \pm 0.6$ | 71.3 |
| ViT-L16$^\dagger$ | $78.2 \pm 0.5$ | $84.6 \pm 0.5$ | $78.0 \pm 0.1$ | $42.7 \pm 1.9$ | 70.9 |
| DeiT$^\dagger$ | $79.3 \pm 0.4$ | $87.8 \pm 0.5$ | $76.6 \pm 0.3$ | $50.0 \pm 0.2$ | 73.4 |
| HViT$^\dagger$ | $79.2 \pm 0.5$ | $89.7 \pm 0.4$ | $80.0 \pm 0.2$ | $51.4 \pm 0.9$ | 75.1 |
| MIRO* | $79.0 \pm 0.0$ | $85.4 \pm 0.4$ | $70.5 \pm 0.4$ | $50.4 \pm 1.1$ | 71.3 |
| MIRO + SWAD* | $79.6 \pm 0.2$ | $88.4 \pm 0.1$ | $72.4 \pm 0.1$ | $52.9 \pm 0.2$ | 73.3 |
| MIRO + SWAG* | $79.9 \pm 0.6$ | $\mathbf{97.4 \pm 0.2}$ | $80.4 \pm 0.2$ | $58.9 \pm 1.3$ | 79.2 |
| MIRO + SWAD + SWAG* | $81.7 \pm 0.1$ | $96.8 \pm 0.2$ | $83.3 \pm 0.1$ | $\mathbf{64.3 \pm 0.3}$ | 81.5 |
| CLIP ViT-B16 | $82.7 \pm 0.3$ | $92.9 \pm 1.9$ | $78.1 \pm 2.1$ | $50.2 \pm 1.7$ | 75.9 |
| DPL | $84.3 \pm 0.4$ | $97.3 \pm 0.2$ | $84.2 \pm 0.2$ | $52.6 \pm 0.6$ | 79.6 |
| **AlignCLIP** | $\mathbf{85.1 \pm 0.2}$ | $97.3 \pm 0.2$ | $\mathbf{86.9 \pm 0.1}$ | $59.5 \pm 0.4$ | $\mathbf{82.2}$ |

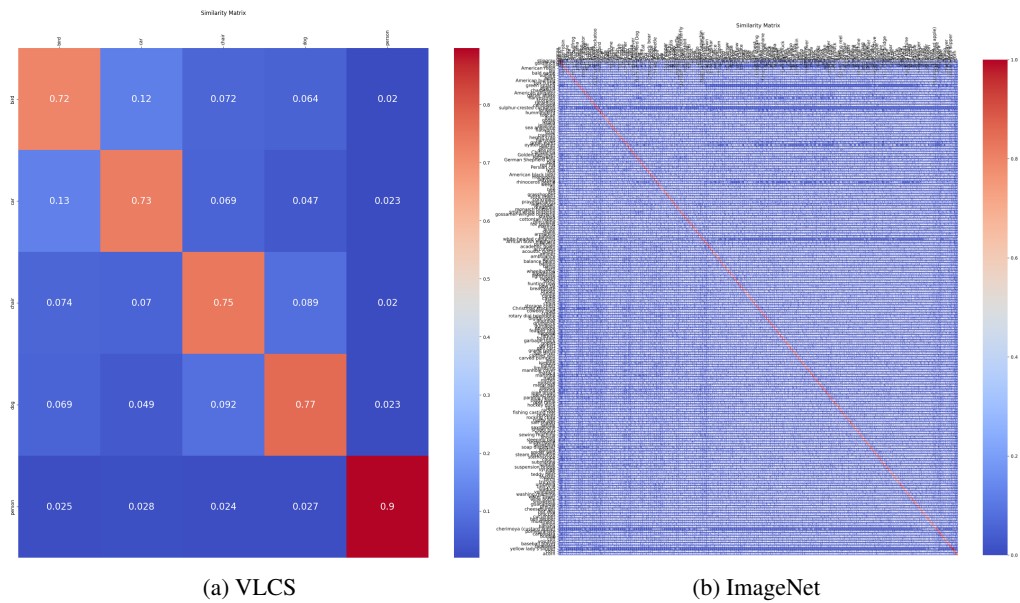

(a) VLCS                    (b) ImageNet

Figure 6: The generated class similarity matrix from the class text embeddings.

## B.3 EXPLORATION OF ALIGNCLIP'S UNIVERSALITY AND VERSATILITY

In this study, our focal point is the open-source version of the CLIP pre-trained model, employing ViT-B/16 as the image encoder. Our exploration extends to assess the universal adaptability of AlignCLIP

across various backbones and different variants of language-vision models. We integrate the CLIP pre-trained model with ResNet-50 as its image encoder (CLIP-RN50), following the methodology of CLIPood (Shu et al., 2023), and incorporate variants of language-vision models such as DeCLIP (Li et al., 2021) and Slip (Mu et al., 2022), utilizing their open-source pre-trained models with ViT-B/32 as the image encoders. For the attention alignment loss, our alignment is specifically targeted at the multi-head attention layer in the image encoder. We meticulously evaluate AlignCLIP on the Office-Home dataset under both domain shift and unseen classes, adhering to the protocol in Section 4.3. The results, illustrated in Figure 5, substantiate that AlignCLIP consistently excels over CLIPood and zero-shot prediction, demonstrating its versatile generalization capability and adaptability across diverse architectures and variants of language-vision models. Table 7 presents the results of applying Empirical Risk Minimization (ERM) using diverse backbone networks on the Domain Generalization (DG) benchmark. The displayed results illustrate that AlignCLIP attains competitive standing, demonstrating comparable or superior performance relative to existing methodologies and a variety of backbone models.

## B.4   ANALYSIS OF THE D-V-STRUCTURE OF ALIGNCLIP

In Figure 7, we present a detailed visual analysis to illustrate the inherent attention alignment challenges within the CLIP model, as seen in the first row. This figure demonstrates that, despite various modifications to the original attention structures (from the third row to the last), the model struggles to correct the attention distributions effectively. These modifications, although diverse, are unable to bring about the desired alignment in attention distributions, highlighting the resilience of the misalignments present in the original model. Even the process of fine-tuning the image encoder in CLIP, depicted in the second row, does not yield substantial alterations or improvements in the attention distribution. This underscores the limitations of fine-tuning as a strategy for addressing attention misalignments in such models, suggesting the need for more robust solutions to mitigate these inherent issues.

To elaborate on the modifications, the Q-Q-Q structure represents a scenario where both $K$ and $V$ vectors are replaced by $Q$, altering the dynamics of the attention mechanism. Similarly, the Q-K-K structure implies a condition where the $V$ vector is substituted by the $K$ vector. These specific alterations, along with others not explicitly mentioned, follow a consistent rule of replacement, aiming to explore the potential of structural modifications in addressing the attention alignment problem. However, as the figure illustrates, these adjustments, while varied, do not succeed in resolving the attention alignment issues inherent to CLIP, indicating a pressing need for more innovative and effective approaches to overcome these challenges in attention-based models.

Figure 8 clarifies the deep implications of synchronizing the $Q$ and $K$ vectors and maintaining the $V$ vector in its designated placeholder. This subtle alteration promotes a more balanced and logical alignment in both similarity and attention distributions, addressing the inherent discrepancies observed in the original configurations. This discovery inspired us to devise a specialized diagonal matrix, strategically engineered to supplant the $Q$ and $K$ vectors. The $\mathbf{D}$-V structure not only preserves the structural and functional integrity of the attention mechanism but also significantly optimizes computational efficiency. By mitigating the computational demands, this method facilitates more streamlined processing, enabling the model to handle more complex computations and larger datasets with ease, thereby broadening the applicability and versatility of the model in diverse computational environments and across varied tasks and domains. Specifically, the accuracy performance between the $\mathbf{D}$-V structure and the V-V-V structure is the same; however, the $\mathbf{D}$-V structure notably improves computational efficiency.

## B.5   CONSTRUCTION AND ANALYSIS OF THE CLASS SIMILARITY MATRIX

**Computation of Similarity.** The process of constructing the class similarity matrix commences with the calculation of similarities between the text embeddings of each class. For each pair of classes, the cosine similarity measure is employed, which assesses the cosine of the angle between two non-zero vectors. Given two vectors $\mathbf{A}$ and $\mathbf{B}$, the cosine similarity is computed as:

$$\text{Cosine Similarity}(\mathbf{A}, \mathbf{B}) = \frac{\mathbf{A} \cdot \mathbf{B}}{\|\mathbf{A}\|_2 \cdot \|\mathbf{B}\|_2},$$

Here, $\mathbf{A} \cdot \mathbf{B}$ denotes the dot product of the vectors, and $\|\mathbf{A}\|_2$ and $\|\mathbf{B}\|_2$ represent the Euclidean norms of the vectors.

**Matrix Construction.** After the cosine similarities between all possible pairs of class text embeddings are computed, we organize the computed values into a $C \times C$ matrix, denoted as $\mathbf{S}$, where $C$ is the number of classes. Each entry $S_{ij}$ in the matrix signifies the cosine similarity between the text embeddings of class $i$ and class $j$.

**Normalization of Similarities.** To normalize the computed similarities and to moderate the influence of extreme values, we apply the softmax function to the raw similarity scores, converting them into probabilities that represent the relative similarities between the classes. The softmax normalization for each entry in the matrix is applied as:

$$S'_{ij} = \frac{\exp(S_{ij})}{\sum_{k=1}^{C} \exp(S_{ik})},$$

Here, $S'_{ij}$ is the normalized similarity between class $i$ and class $j$, and the denominator is the sum of the exponentiated similarity scores of class $i$ with all other classes.

**Utilization of the Class Similarity Matrix.** The resultant normalized matrix, $\mathbf{S}$, encapsulates the semantic relationships between the classes, providing insights into the inherent semantic structure and enabling more informed and coherent predictions by leveraging the revealed inter-class relationships. As depicted in Figure 6, the constructed similarity matrix effectively reflects the inherent similarity between classes, revealing nuanced relationships that are crucial for model generalization. For instance, within the VLCS dataset, the similarity between the classes 'bird' and 'car' is notably higher than the similarity between 'bird' and 'chair'. This observation underscores the capability of class text embeddings to accurately capture the true semantic similarity between classes, a feature that is ingrained during the language-vision pre-training phase. The accurate representation of inter-class similarities is instrumental in enhancing the model's ability to generalize across varied classes, thereby contributing to its overall performance in diverse application scenarios. For plotting the attention maps, we calculate the similarity distance between the features of image tokens and the transposed text features, applying L2 normalization. This approach allows for a direct and meaningful comparison of attention distributions.

To justify that the CLIP pretrained text encoder can capture a hierarchy of semantic relationships, here are three key points supported by relevant research and findings: (1) CLIP has been trained on a massive scale, with 400 million pre-training image-text pairs. This extensive dataset ensures that CLIP's text encoder can effectively capture subtle semantic differences in text, as it has been exposed to a wide variety of linguistic contexts and nuances; (2) Previous studies have demonstrated that the text encoder of CLIP inherently possesses a fine-grained understanding of semantic hierarchies [1][2]; (3) Research [3] shows that the CLIP text encoder is not only capable of understanding phrases but can also outperform popular language models like BERT when prompted appropriately. This indicates its strong ability in various text-understanding tasks, demonstrating its robustness and effectiveness as a backbone in multiple applications.

**Significance of Semantic Relationships.** The ability to quantify and understand the semantic relationships between classes through the class similarity matrix is of paramount importance. It not only aids in the model's predictive accuracy but also provides valuable insights into the underlying structure of the data. By leveraging the nuanced relationships captured in the matrix, models can make more semantically coherent predictions, especially in scenarios where distinctions between classes are subtle and complex. This enhanced semantic understanding, derived from the class similarity matrix, is a crucial component in optimizing the model's adaptability and generalization capabilities in open-world environments. The parameter $\epsilon$ in equation 8 plays a pivotal role in our semantic label smoothing technique, balancing the influence of the original and smoothed labels. The experimental results in Table 8 demonstrate that $\epsilon$ is not too sensitive to set and semantic label smoothing achieves obvious performance gain (compared to $\epsilon = 1$).

**Pooling Technique.** One may argue that using other pooling techniques after the last layer (e.g. average pooling) instead of using the class token can achieve better performance in the CLIP fine-tuning phase. To investigate this, we conducted additional experiments employing average pooling (AP) across visual tokens, as opposed to using class tokens (CT). The results, detailed in Table 8, reveal that AP yields lower performance compared to CT. This outcome suggests that while the

Table 8: The quantitative results of $\epsilon$ on the VLCS dataset.

| DOMAIN | 1 | 0.9 | 0.8 | 0.7 | 0.6 |
|--------|-----|------|------|------|------|
| LABELME | 67.4±0.2 | 68.3±0.2 | 68.3±0.1 | 67.9±0.1 | 67.9±0.2 |
| VOC2007 | 88.8±0.1 | 89.9±0.2 | 89.8±0.1 | 89.7±0.2 | 89.5±0.2 |

Table 9: The quantitative results for average pooling (AP) and class tokens (CT) by fine-tuning original CLIP on the VLCS dataset.

| METHOD | CALTECH101 | LABELME | SUN09 | VOC2007 | AVG |
|--------|-----------|---------|-------|---------|-----|
| CLIP-AP | 99.6±0.1 | 65.4±0.2 | 77.3±0.2 | 85.6±0.3 | 82.0 |
| CLIP-CT | 99.3±0.1 | 66.4±0.4 | 80.3±0.2 | 88.7±0.3 | 83.7 |

class token effectively represents images in training, other tokens receive less emphasis. Moreover, the limited scale of downstream datasets in fine-tuning may not sufficiently optimize all tokens, highlighting the practicality of our original approach.

## C    DISCUSSION

### C.1    COMPUTATIONAL COMPLEXITY

Our method, while employing a Siamese-type architecture for fine-tuning, introduces only a minimal increase in training computational complexity compared to CLIPood and does not increase the number of parameters or inference speed. The reasons are as follows.

**Inference speed:** The Siamese architecture is utilized only during the training phase. At inference, the speed is identical to that of CLIPood. For instance, with a batch size of 36, the inference speed for both is 1.973 seconds. This parity in speed underscores the efficiency of our method during practical applications.

**Parameter Sharing:** The two branches of the Siamese architecture share parameters, meaning there is no increase in the total number of model parameters. This design choice ensures that our model remains lightweight and scalable.

**Training Speed:** While there is a slight decrease in training speed, it is marginal. On a Tesla V100 GPU, with a batch size of 36, the training speed for CLIPood is 0.19 seconds per batch, compared to 0.21 seconds for AlignCLIP, a minor difference of 0.02 seconds. It's important to note that our D-V attention structure significantly reduces the GFLOPS and training speed (0.25 seconds) compared to the V-V-V attention structure.

**Comparison with Prompt Tuning Methods:** When compared to prompt tuning methods like MaPLe (Khattak et al., 2023) and CoOP (Zhou et al., 2022b), our training time is not directly comparable as our method, similar to CLIPood (Shu et al., 2023), is a new parameter fine-tuning approach. However, roughly speaking, MaPLe and CoOP are faster in training as they do not require fine-tuning the parameters of the image encoder. The inference speeds are almost identical. CoCoOP (Zhou et al., 2022a) has a slower training speed due to its use of image representations to calculate prompts, necessitating smaller batch sizes to accommodate GPU memory constraints.

### C.2    ATTENTION MISALIGNMENT

Two seminal studies (Li et al., 2022b; 2023) have demonstrated that CLIP models exhibit a preference for background regions over foreground elements, a tendency that deviates from human perception. These papers provide substantial evidence that CLIP's attention mechanisms often prioritize background information, contradicting what is typically expected in human-centered image understanding.

In our work, we present clear evidence through empirical analysis (see Figure 1, Supplementary Figures 7 and 8). These figures illustrate the tendency of CLIP models to disproportionately focus on background elements. Our analysis goes beyond mere observation by offering a methodological approach to recalibrate the attention mechanism, thereby refocusing it on salient objects rather than the background. The training dataset of CLIP, consisting of billions of image-text pairs, often reflects images as effects of certain concepts (Y), akin to the mental visualization preceding its depiction. The accompanying text, however, encompasses a broader semantic range, capturing not just the effect (content) but also the cause (background context). Post-training, CLIP models are proficient in identifying these causal relationships. Our model leverages this capability in downstream tasks, effectively discerning and utilizing these causal connections, which further validates the focus of the D-V attention map in attention learning.

## C.3 D-V Structure

Regarding the use of the D-V structure's attention map as a target distribution for attention learning, we would like to offer a multi-faceted discussion that supports the validity of our approach. Firstly, the issue of attention misalignment in CLIP models has been empirically demonstrated in the work of CLIP Surgery Li et al. (2023). It has been shown that the attention mechanism during training, particularly the interactions between Query (Q) and Key (K), does not always capture the semantic essence of the tokens, which is more accurately represented by the Value (V). Based on this finding, the CLIP Surgery paper proposed a modification of the Q-K-V attention structure that replaces the Query and Key vectors with the V vector. Empirical studies in this paper demonstrate that the V-V-V structure can obtain ideal and reasonable predictive attention maps (please refer to this paper). Inspired by the V-V-V structure, we propose the D-V attention structure that replaces the product of Key and Query in self-attention with a diagonal matrix. Our D-V structure prioritizes this semantic essence of V vectors, therefore providing a more aligned target for attention learning. Secondly, standard self-attention weights the sum of value vectors by the matching scores of Q and K. In our work, the product of Q and K in self-attention is replaced with a diagonal matrix, meaning that the new representation of each input element relies solely on its own information, rather than the interrelation with other elements. This change encourages the model to focus more on the salient targets rather than the background when generating attention maps. By shifting the focus from inter-token relationships to the intrinsic semantic information of each token, our approach ensures that the attention mechanism is more closely aligned with the inherent content of the input, thus serving as an ideal target for attention learning. We believe that this modification is a significant step towards more accurately modeling visual attention in line with human cognitive patterns.

## C.4 Limitation and Future Work

Through detailed testing across different distribution shifts and unseen categories, AlignCLIP has consistently outperformed zero-shot pre-trained models and current generalization methods, showcasing its reliability and adaptability in various open-world situations. AlignCLIP is carefully designed, combining three key elements: Attention Alignment Loss, Semantic Label Smoothing, and **D**-V Structure. Each element is crucial in boosting the model's performance and its ability to adapt while keeping additional computational needs low.

However, AlignCLIP does have its challenges. One major limitation is its inefficiency in adjusting all the parameters of the image encoder during fine-tuning, which could limit its use in resource-restricted settings. Also, while the addition of Attention Alignment Loss improves the attention distribution, it doesn't fully correct it. The minor adjustments made to the pretraining parameters aren't enough to bring about a complete change in all the parameters, indicating there's still room for improvement in optimizing attention distributions.

Given these challenges, future research could look into developing new pretraining methods that can more effectively correct attention distributions. Progress in this area could lead to models with better adaptability and understandability, potentially setting new benchmarks in OOD generalization approaches.

## C.5 About the difference from CLIP Surgery (Li et al., 2023)

**The research topics are different.** The research topic of CLIP Surgery is model interpretability while our research topic is out-of-distribution generalization and out-of-distribution detection. Specifically, CLIP Surgery concerns neural network interpretability while our work addresses the challenge of creating stable representations in VLP models for adaptability in open-world environments, particularly dealing with domain shifts and unseen classes.

**Our work has the V-V-V attention structure advance.** As for the modified structure of attention, our work provides a new insight for the community. Specifically, the CLIP surgery paper only points out only the V-V-V structure is working, but our work points out that not only the V-V-V structure can work, but also works as long as Q-K is replaced by an equivalent structure (such as K-K, Q-Q, or diagonal matrix), as shown in Figure 8. This insight opens opportunities for future studies to design better model interpretability methods and even to guide the design of vision-language-pertaining strategies to avoid the attention misalignment problem.

**Our work has more methodology contributions.** It is important to highlight that our contributions extend beyond the adoption of the D-V structure. While we acknowledge the foundational work done by Li et al. (2023), our approach diverges in addressing the misalignment problem rather than pointing out it only. These include the AlignCLIP which includes a new optimization objective (Attention Alignment Loss) and a regularization technique (Semantic Label Smoothing). Our work thus provides a novel perspective and contributes to the field by enhancing the stability and adaptability of representations in VLP models, ensuring better performance across varied datasets and out-of-distribution contexts.

**D-V structure has computation advance.** Our decision was primarily driven by the goal of achieving greater computational efficiency. To elaborate, we have performed a detailed GFLOPS computation for both structures in the context of a multi-head attention mechanism. For the V-V-V structure, the FLOPs per head are calculated as $2 \times N \times (d_v \times d_{model} + d_v \times d_{model})$, where $N$ represents the sequence length, $d_v$ is the dimension of the value, and $d_{model}$ is the model dimension. In contrast, for the D-V structure, the FLOPs per head simplifies to $N \times d_v$. Applying these calculations to a vision transformer model with a 12-head attention mechanism, we found that the GFLOPS for the V-V-V structure amounts to 0.46478, whereas for the D-V structure, it is significantly lower at just 0.0003. This stark contrast indicates that the D-V structure reduces the computational demand by approximately 1549 times compared to the V-V-V structure. Such a substantial reduction in GFLOPS directly translates into markedly faster processing times and reduced resource consumption. This efficiency makes our approach particularly advantageous for real-world applications, especially in contexts where computational resources are constrained or limited. We believe that these calculations demonstrate the superiority of the D-V structure in terms of computational efficiency, thereby justifying our design choice and reinforcing the innovative nature of our work.

**Quantitative results for V-V-V and D-V structure.** Regarding the quantitative performance comparison, we have conducted additional experiments to provide a direct comparison between the D-V and V-V-V structures. As shown in Table 10, we observed that the D-V structure marginally outperforms the V-V-V structure in specific tasks. We have included these results in Table xx in the revised manuscript.

Table 10: The quantitative results for V-V-V and D-V structure on the VLCS dataset.

| Method | Caltech101 | LabelMe | SUN09 | VOC2007 | Avg |
|--------|-----------|---------|-------|---------|-----|
| V-V-V | 99.6±0.2 | 68.7±0.2 | 81.1±0.1 | 90.07±0.2 | 84.9 |
| D-V | 100±0.0 | 68.9±0.3 | 80.9±0.1 | 90.08±0.2 | 85.1 |

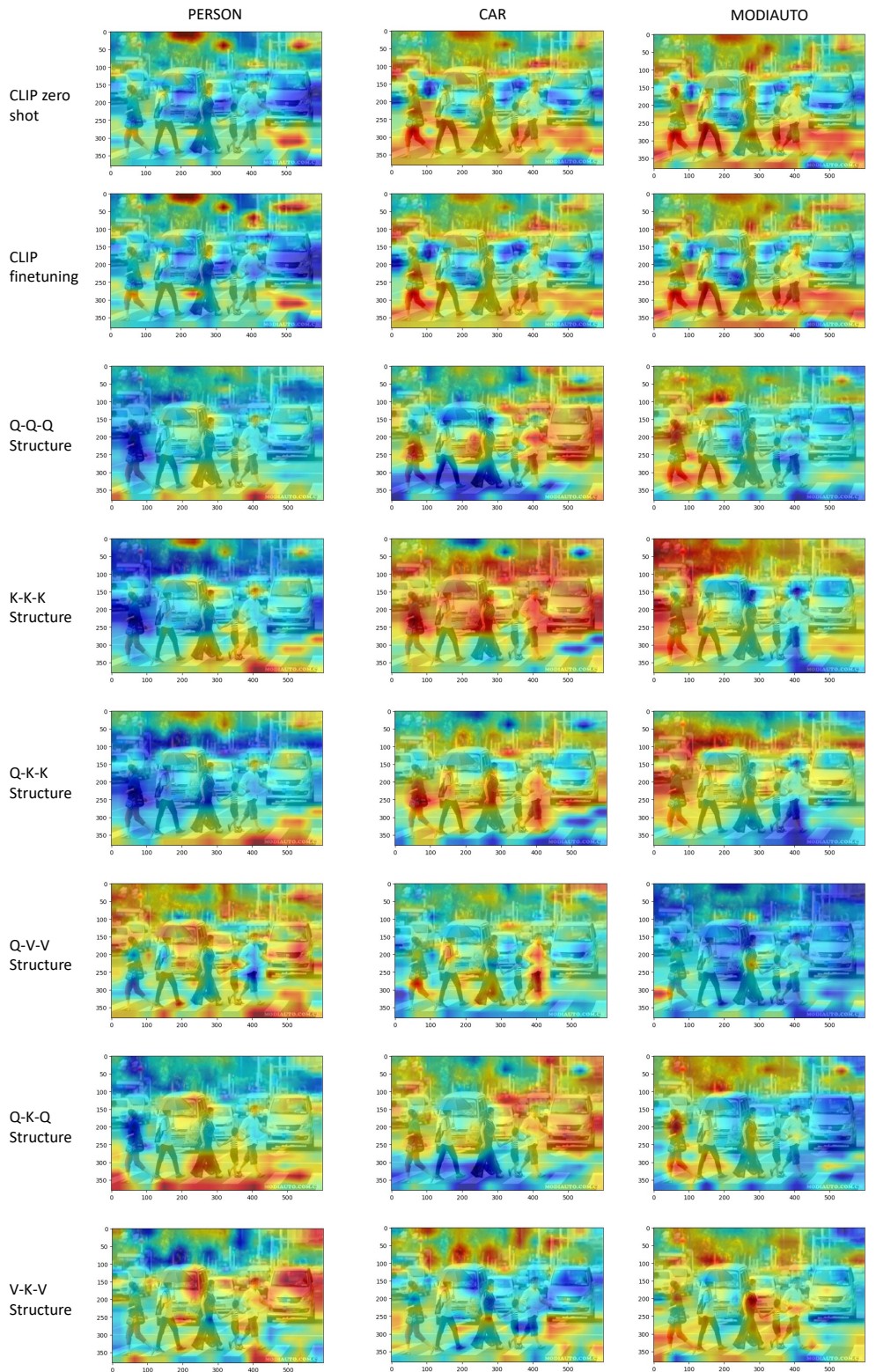

Figure 7: Illustration depicting the inherent attention alignment issues in CLIP (first row). Subsequent adjustments to the original attention structures, spanning from the third row to the last, are unable to rectify the attention distributions effectively.

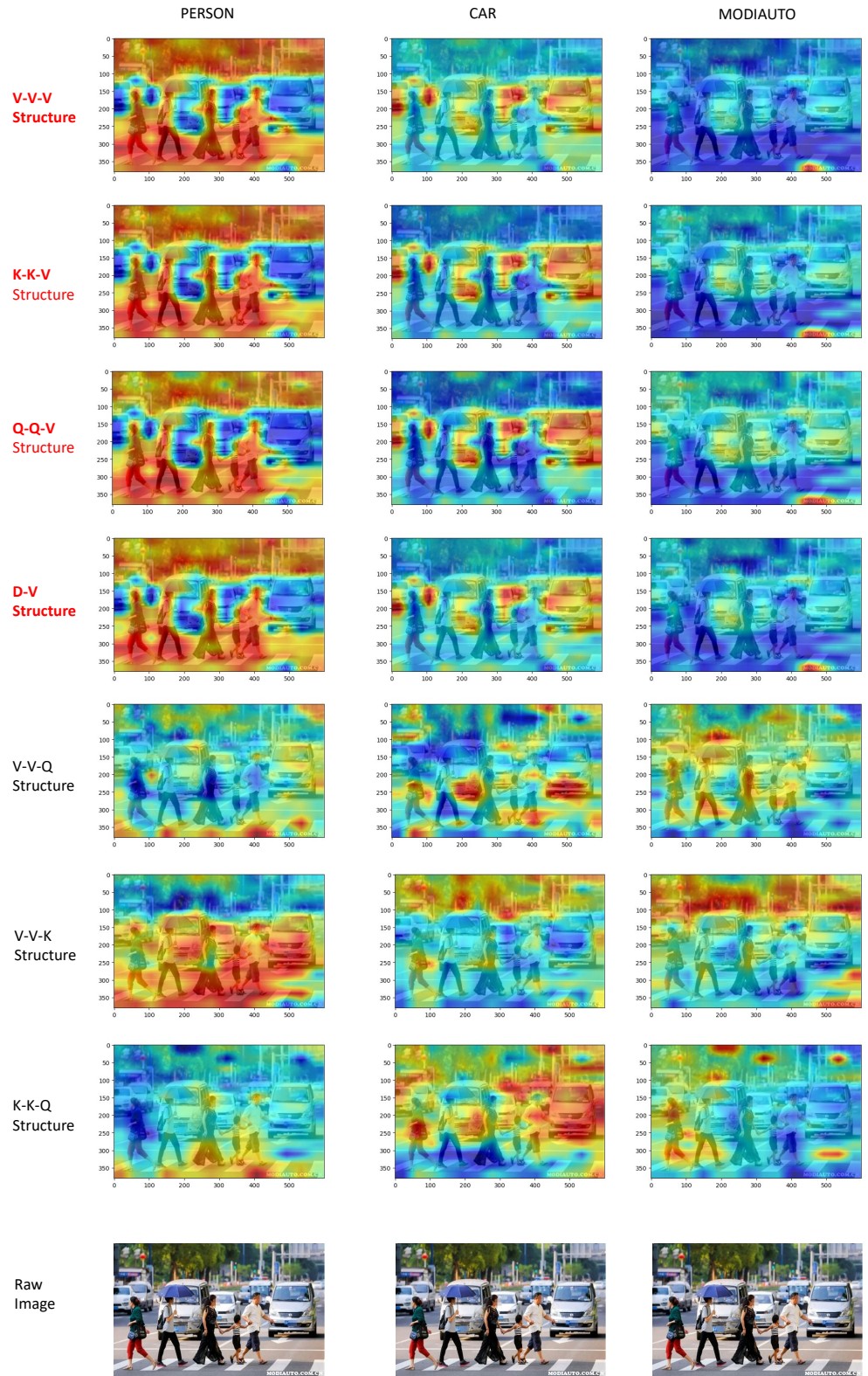

Figure 8: This figure illustrates the impact of equating the Q and K vectors while retaining the V vector in its original placeholder, a modification that results in more coherent similarity and attention distributions. 'MODIAUTO' refers to the text contained in the images.

