# OpenReview forum: "AlignCLIP: Enhancing Stable Representations in Vision-Language Pretraining Models through Attention and Prediction Alignment"
_ICLR.cc/2024/Conference — Submitted to ICLR 2024_

### Official Review · Reviewer_TgXT · 2023-10-30

**Soundness:** 3 good
**Presentation:** 3 good
**Contribution:** 3 good
**Rating:** 5
**Confidence:** 4

**Summary:**

To address the attention misalignment in CLIP, this paper introduces AlignCLIP, that want to finetune CLIP by designing two regularization loss: 1)Attention alignment loss that aims to regularize the attention distributions at all L encoder layers with the similarity matrix of the label and image patch embeddings. And 2) Semantic label smoothing that wants to explore relationships between different classes. Both numerical results on domain-shift and unseen tasks and visualization analysis show the superior performance of the proposed model. A series of ablation results are further provided to show the impact of the introduced two modules.

**Strengths:**

1) One of the main contributions of this paper is to address the attention misalignment in CLIP, which is a challenge issue in the community. The proposed method that aligns the attention distribution with the semantic similarity of label and patch embeddings sounds interesting.

2) The organization of this article is logical, it's well-written, and the figures are clear, making it easy for readers to understand the author's idea.

3) Extensive comparisons on two tasks show the improvements of the proposed. The visualization well demonstrates the motivation, and the ablation studies analyze the introduced two modules.

**Weaknesses:**

1) One of the main concerns comes from Eq.(6) in the manuscript. The proposed AAL views the $G_{D-V}$ as the groundtruth attention distribution and aligns $A_c$ and $G$ with $G_{D-V}$. Given the fact the value vectors is problematic and $G_{D-V}$  is obtained based on the V vector at the D-V branch, I do not know why $G_{D-V}$ can be viewed as the groundtruth guidance.

2) The introduced AAL and SLS seem to be independent, and Table (1-4) show that the improvements is slight.  I am not sure that applying AAL or SLS separately still provides a stable improvement.

**Questions:**

1) $G_{D-V}$ in Eq.(6) is not defined or described.

2) Does $\theta$ in Algorithm 1 denote the all CLIP parameters or only the image encoder's parameters?

3) I found that there are newly introduced parameters in the new block (Figure 2) and whether they are fixed or need to be learned.

---

> ### Author Response · Authors · 2023-11-20
> **Response to W1 of Reviewer TgXT**
>
> **W1: One of the main concerns comes from Eq.(6) in the manuscript. The proposed AAL views the $G_{D-V}$ as the groundtruth attention distribution and aligns $A_c$ and with $G_{D-V}$. Given the fact the value vectors is problematic and $G_{D-V}$ is obtained based on the V vector at the D-V branch, I do not know why $G_{D-V}$ can be viewed as the groundtruth guidance.**
>
> **Reply:** Thank you for your valuable comments regarding the use of the D-V structure's attention map as a target distribution for attention learning. We would like to offer a multi-faceted response that supports the validity of our approach.
>
> 1. **Empirical foundation from related work:** The issue of attention misalignment in CLIP models has been empirically demonstrated in the work of CLIP Surgery [1]. It has been shown that the attention mechanism during training, particularly the interactions between Query (Q) and Key (K), does not always capture the semantic essence of the tokens, which is more accurately represented by the Values (V). Based on this finding, the CLIP Surgery paper [1] proposed a modification of the Q-K-V attention structure that replaces the Query and Key vectors with the V vector. Empirical studies in this paper demonstrate that the V-V-V structure can obtain ideal and reasonable predictive attention maps (please refer to the CLIP Surgery paper).
>
>     In this paper, the purpose of proposing the D-V structure is to explain that not only the V-V-V structure proposed in the CLIP surgery paper can work, but also works as long as Q-K is replaced by an equivalent structure (such as K-K, Q-Q, or diagonal matrix), as shown in Figure 8 at the Appendix. Based on this finding, we propose the D-V attention structure that replaces the product of Key and Query in self-attention with a diagonal matrix. The advantage is that has a small computation complexity. we found that the GFLOPS for the V-V-V structure amounts to 0.46478, whereas for the D-V structure, it is significantly lower at just 0.0003. Our D-V structure prioritizes this semantic essence of V vectors, therefore providing a more aligned target for attention learning.
>
>     [1] Yi Li, Hualiang Wang, Yiqun Duan, and Xiaomeng Li. CLIP Surgery for Better Explainability with Enhancement in Open-Vocabulary Tasks. arXiv, 2023.
>
> 2. **Self-attention mechanism in transformers:** In the Transformer model, Self-Attention is achieved through three vectors: Query (Q), Key (K), and Value (V). For each input element (image patch in ViT), Q, K, and V are computed. Standard Self-Attention weights the sum of Values (V) by the matching scores of Queries (Q) and Keys (K). In our work, the product of Key and Query in Self-Attention is replaced with a diagonal matrix, meaning that the new representation of each input element relies solely on its own information, rather than the interrelation with other elements. This change encourages the model to focus more on the salient targets rather than the background when generating attention maps. By shifting the focus from inter-token relationships to the intrinsic semantic information of each token, our approach ensures that the attention mechanism is more closely aligned with the inherent content of the input, thus serving as an ideal target for attention learning. We believe that this modification is a significant step towards more accurately modeling visual attention in line with human cognitive patterns. We thank the other reviewers for appreciating our contribution to attention alignment. Reviewer TgXT points out that "The proposed method that aligns the attention distribution with the semantic similarity of label and patch embeddings sounds interesting."
>
> `Revision: We have further elaborated on this argument on Page 23, Section of C.3 in the manuscript, providing additional empirical evidence and mechanism analysis to strengthen our claim.`

---

> > ### Author Response · Authors · 2023-11-20
> > **Response to W2 of Reviewer TgXT**
> >
> > **W2: The introduced AAL and SLS seem to be independent, and Table (1-4) show that the improvements is slight. I am not sure that applying AAL or SLS separately still provides a stable improvement.**
> >
> > **Reply:** Thank you for your feedback on the motivations behind our attention alignment and label smoothing techniques. Firstly, we appreciate the opportunity to clarify how these are indeed cohesively aligned:
> >
> > 1. **Unified Motivation:** Both techniques aim to address the misalignment in CLIP models by leveraging textual information, ensuring consistency between visual and textual modalities. The unifying motivation behind both Attention Alignment Loss (AAL) and Semantic Label Smoothing (SLS) is to enhance the adaptability and generalization of CLIP models in open-world settings, where both domain shifts and unseen classes are present. While AAL focuses on optimizing the attention mechanism to better focus on relevant features, SLS aims to refine the prediction mechanism to be more semantically accurate and less prone to misclassification. Together, they address different aspects of the model’s performance but work towards the same goal of improving stability and reliability in complex, real-world scenarios.
> >
> > 2. **Common Goal of Stable Representations:** Both attention alignment and semantic label smoothing contribute to the overarching goal of achieving stable representations in Vision-Language Pretraining (VLP) models. This stability is crucial for handling challenges like domain shifts and recognizing unseen classes in open-world environments. Therefore, their shared objective is to improve the representational stability and generalization of CLIP models, crucial for performance in open-world settings with domain and class variations.
> >
> > 3. **Textual Information Utilization:** Textual information is integral to both methods—fine-tuning attention layers and refining output distribution—demonstrating their conceptual harmony.
> >
> > 4. **Stable Improvement from Both Modules:** We have revised the bar charts in Figure 3 on Page 9 in the revised manuscript to demonstrate the stable improvement. The updated chart better highlights the incremental improvements provided by AAL and SLS. Despite appearing modest, these gains are statistically significant and essential for the robustness of our model. AAL improves accuracy by approximately 1.0\% in the VLCS dataset and 2.8\% in the Office-Home dataset. SLS alone enhances accuracy by 0.7\% in VLCS. Importantly, their combined use results in a consistent and notable improvement (1.4\%) in the Domain+Class scenario, underscoring our model's effectiveness in handling real-world domain shifts. The revised figure distinctly contrasts SLS with standard label smoothing. SLS shows improvements of 3.21\%, 2.29\%, and 0.89\% in LabelMe, SUN, and VOC2007 datasets, respectively. These enhancements affirm the capability of SLS to utilize semantic class relationships effectively, proving particularly beneficial in complex scenarios marked by distribution shifts and the presence of open classes.
> >
> > `Revision: 1. We have enhanced the detailed analysis of how these modules interlink on Page 8, Section 4.4 in the revised manuscript, providing additional evidence of their synergistic relationship. We hope this addresses your concerns and better illustrates the aligned motivations of our contributions. 2. We have revised the bar charts in Figure 3 on Page 9 in the revised manuscript to include annotations that distinctly showcase the substantial contributions of AAL and SLS in scenarios involving domain shift and open classes.`

---

> > > ### Author Response · Authors · 2023-11-20
> > > **Response to Q1-Q3 of Reviewer TgXT**
> > >
> > > **Q1: $G_{D-V}$ in Eq.(6) is not defined or described.**
> > >
> > > **Reply:** Thank you for your inquiry regarding the "D-V" structure. Let me clarify this concept:
> > >
> > > The D-V structure is an innovative attention structure we introduced. In conventional Q-K-V attention blocks, attention weights are computed as $\text{Softmax}\left(\frac{QK^T}{\sqrt{d_k}}\right)$, where $Q$, $K$, and $V$ denote the query, key, and value vectors, respectively, and $d_k$ is the dimension of the keys. In our D-V structure, we modify this mechanism by replacing the softmax-normalized product of $Q$ and $K^T$ with a diagonal matrix. This approach emphasizes individual token features, utilizing the value vector $V$ more directly. Furthermore, $G_{D-V}(X_i)$ denotes the normalized attention distribution obtained by calculating the cosine similarity distance between image token features, using the D-V structure, and transposed text features.
> > >
> > > `Revision: In the revised manuscript, we have provided a clear definition of the D-V structure and $G_{D-V}$ on Page 5, Section 3.2, ensuring comprehensive understanding. We explicitly explain that $G_{D-V}$ represents the modified attention guidance derived from our D-V structure.`
> > >
> > > **Q2: Does $\theta$ in Algorithm 1 denote all CLIP parameters or only the image encoder's parameters?**
> > >
> > > **Reply:** Thanks for your question. $\theta$ in Algorithm 1 denotes the image encoder's parameters.
> > >
> > > `Revision: We have revised Algorithm 1 on Page 6 in the revised manuscript. It is more clear now.`
> > >
> > > **Q3: I found that there are newly introduced parameters in the new block (Figure 2) and whether they are fixed or need to be learned.**
> > >
> > > **Reply:** Thanks for your valuable question. There is a potential confusion that the two branches of the Siamese architecture share parameters, meaning there is no increase in the total number of model parameters. This design choice ensures that our model remains lightweight and scalable.
> > >
> > > `Revision: We have added clarification to make it clearer to avoid potential confusion in Figure 2 of the revised manuscript.`

---

> > > > ### Comment · Reviewer_TgXT · 2023-11-22
> > > > **Thank you for the response**
> > > >
> > > > Based on the responses of authors, and comments of other reviewers, I decided to keep my original rating.

---

> > > > > ### Author Response · Authors · 2023-11-23
> > > > >
> > > > > Dear Reviewer TgXT,
> > > > >
> > > > > Thank you for your valuable feedback on our manuscript. We have carefully considered your comments and made the following key revisions to address the points you raised:
> > > > >
> > > > > 1. On Page 23, Section C.3, we have elaborated our argument regarding the use of the D-V structure's attention map as a target distribution for attention learning by providing additional empirical evidence and a deeper mechanism analysis to solidify our claim.
> > > > >
> > > > > 2. On Page 8, Section 4.4, we enhanced the analysis detailing the interlinking of the modules, adding evidence of their synergistic relationship to clarify the aligned motivations of our contributions.
> > > > >
> > > > > 3. In Figure 3 on Page 9, we revised the bar charts to include annotations, offering a clearer representation of the significant impacts of AAL and SLS in scenarios of domain shift and open classes.
> > > > >
> > > > > 4. We provided a comprehensive definition and explanation of the D-V structure and $G_{D-V}$ on Page 5, Section 3.2, for a better understanding of these concepts.
> > > > >
> > > > > 5. Algorithm 1 on Page 6 has been revised for greater clarity.
> > > > >
> > > > > 6. We have made additions and clarifications in Figure 2 to avoid any potential confusion.
> > > > >
> > > > > We believe these revisions significantly enhance the manuscript and address your concerns. We hope that these changes meet your approval and look forward to any further suggestions you may have.

---

### Official Review · Reviewer_v1vj · 2023-10-31

**Soundness:** 3 good
**Presentation:** 3 good
**Contribution:** 2 fair
**Rating:** 5
**Confidence:** 4

**Summary:**

This paper proposes a method to enhance image and text alignment for a pretrained CLIP model by aligning the attention and smooth label prediction. The motivation for the attention alignment is from the bad attention map produced by the original CLIP model. Labeling smooth is motivated by the fact that not all incorrect classes should contribute equally. The experiments are conducted on two scenarios: domain shift and unseen class.

**Strengths:**

This paper is well motivated, despite some unclear explanation points. (see below). Particularly, the plotted figures in the method part make high-level understanding easier.

The experiments are extensively conducted, covering a number of different datasets in two scenarios. The ablation study is generally satisfactory, despite missing one key parameter analysis. (see below)

**Weaknesses:**

For the two technical contributions on attention alignment and label smoothing, I feel they are disjointed. The motivations are not well aligned as well.



In abstract, it is stated that label smoothing aims to persevere prediction hierarchy, and use the “dog” and “wolf” example. This is interesting. However, I don’t see any justification on this point, except few picked examples. Moreover, whether the pretrained text encoder can capture such a “hierarchy” also needs justification.



Parameter $\epsilon$ is important, where its analysis is missing.



Another is that although the proposed method outperforms the baseline for the most cases, its improvement over CLIPood is marginal.

**Questions:**

I was not clear about how the inference works. If Eq 4 is involved in inference stage, then one major concern is that the proposed method might not be easy to scale up in practice. Due to the requirement of precomputing text embedding  $T_y$ in Eq. 4 for each visual token similarity computation, this “cross-attention” based image embedding will have the same number of versions as the number of classes. This will limit the application when the number of classes is large. On the contrary, for CLIP, image embedding is not conditioned on text.



I am not sure how the CLIP zero-shot and finetune attention maps (e.g. figure 1 (b)) plotted. Please explain. Considering CLIP training takes the last token as the image representation trained with a casual mask, its individual tokens are intentionally designed for such attention alignment. For a more convincing comparison, in CLIP finetune, the authors might want to use other pooling techniques after the last layers (e.g. average pooling) instead of using the last tokens. In this case, each visual token will be enforced to have better representation.



It is unclear to me what does “$D-V$” structure mean. How $G_{D-V}(X_i)$ is defined.

---

> ### Author Response · Authors · 2023-11-20
> **Response to W1 of Reviewer v1vj**
>
> **W1: For the two technical contributions on attention alignment and label smoothing, I feel they are disjointed. The motivations are not well aligned as well.**
>
> **Reply:** Thank you for your feedback on the motivations behind our attention alignment and label smoothing techniques. Firstly, we appreciate the opportunity to clarify how these are indeed cohesively aligned:
>
> 1. **Unified Motivation:** Both techniques aim to address the misalignment in CLIP models by leveraging textual information, ensuring consistency between visual and textual modalities. The unifying motivation behind both Attention Alignment Loss (AAL) and Semantic Label Smoothing (SLS) is to enhance the adaptability and generalization of CLIP models in open-world settings, where both domain shifts and unseen classes are present. While AAL focuses on optimizing the attention mechanism to better focus on relevant features, SLS aims to refine the prediction mechanism to be more semantically accurate and less prone to misclassification. Together, they address different aspects of the model’s performance but work towards the same goal of improving stability and reliability in complex, real-world scenarios.
>
> 2. **Common Goal of Stable Representations:** Both attention alignment and semantic label smoothing contribute to the overarching goal of achieving stable representations in Vision-Language Pretraining (VLP) models. This stability is crucial for handling challenges like domain shifts and recognizing unseen classes in open-world environments. Therefore, their shared objective is to improve the representational stability and generalization of CLIP models, crucial for performance in open-world settings with domain and class variations.
>
> 3. **Textual Information Utilization:** Textual information is integral to both methods—fine-tuning attention layers and refining output distribution—demonstrating their conceptual harmony.
>
> 4. **Stable Improvement from Both Modules:** We have revised the bar charts in Figure 3 on Page 9 in the revised manuscript to demonstrate the stable improvement. The updated chart better highlights the incremental improvements provided by AAL and SLS. Despite appearing modest, these gains are statistically significant and essential for the robustness of our model. AAL improves accuracy by approximately 1.0\% in the VLCS dataset and 2.8\% in the Office-Home dataset. SLS alone enhances accuracy by 0.7\% in VLCS. Importantly, their combined use results in a consistent and notable improvement (1.4\%) in the Domain+Class scenario, underscoring our model's effectiveness in handling real-world domain shifts. The revised figure distinctly contrasts SLS with standard label smoothing. SLS shows improvements of 3.21\%, 2.29\%, and 0.89\% in LabelMe, SUN, and VOC2007 datasets, respectively. These enhancements affirm the capability of SLS to utilize semantic class relationships effectively, proving particularly beneficial in complex scenarios marked by distribution shifts and the presence of open classes.
>
> `Revision: 1. We have enhanced the detailed analysis of how these modules interlink on Page 8, Section 4.4 in the revised manuscript, providing additional evidence of their synergistic relationship. We hope this addresses your concerns and better illustrates the aligned motivations of our contributions. 2. We have revised the bar charts in Figure 3 on Page 9 in the revised manuscript to include annotations that distinctly showcase the substantial contributions of AAL and SLS in scenarios involving domain shift and open classes.`

---

> > ### Author Response · Authors · 2023-11-20
> > **Response to W2 of Reviewer v1vj**
> >
> > **W2: In abstract, it is stated that label smoothing aims to persevere prediction hierarchy, and use the “dog” and “wolf” example. This is interesting. However, I don’t see any justification on this point, except few picked examples. Moreover, whether the pretrained text encoder can capture such a “hierarchy” also needs justification.**
> >
> > **Reply:** We greatly appreciate your interest in our work, particularly your observations regarding our application of semantic label smoothing to preserve prediction hierarchy, as exemplified by the “dog” and “wolf” analogy in the abstract. This significantly improves the representation stability in open-world scenarios. We offer the following justifications:
> >
> > 1. **Empirical Evidence Supporting SLS:** In addition to the conceptual explanation, we have provided empirical evidence supporting the efficacy of SLS. As illustrated in Figure 4d of our paper, SLS shows improvements of 3.21\%, 2.29\%, and 0.89\% in the LabelMe, SUN, and VOC2007 datasets, respectively. These enhancements affirm the capability of SLS to utilize semantic class relationships effectively. The improvement is particularly noticeable in complex scenarios marked by distribution shifts and the presence of open classes. These results are not just statistically significant but also demonstrate the practical impact of SLS in diverse and challenging real-world conditions.
> >
> > 2. **Explanation of SLS Mechanism:** To further clarify, SLS aligns the model’s predicted category distribution with the semantic relationships among classes, which are established during the language-vision pre-training phase. Our approach to label smoothing is not arbitrary. It is specifically designed to consider the semantic proximity between classes, which is inherently hierarchical. For example, "dog" and "wolf" share a closer relationship with each other than either does with "fish." Our method applies a smoother distribution of probabilities that acknowledges these nuances, allocating a higher probability to "wolf" when predicting "dog" and vice versa.
> >
> > 3. **CLIP Text Encoder's Hierarchical Capture:** In the original paper, as shown in Figure 6 of the appendix, we have illustrated the generated class similarity matrices from the class text embeddings outputted by the CLIP pretrained text encoder. These two class similarity matrices can demonstrate that the CLIP text encoder can capture the hierarchical relationships between classes. Further, to justify that the CLIP pretrained text encoder can capture a "hierarchy" of semantic relationships, here are three key points supported by relevant research and findings: (1) CLIP has been trained on a massive scale, with 400 million pre-training image-text pairs. This extensive dataset ensures that CLIP's text encoder can effectively capture subtle semantic differences in text, as it has been exposed to a wide variety of linguistic contexts and nuances; (2) Previous studies have demonstrated that the text encoder of CLIP inherently possesses a fine-grained understanding of semantic hierarchies [1][2]; (3) Research [3] shows that the CLIP text encoder is not only capable of understanding phrases but can also outperform popular language models like BERT when prompted appropriately. This indicates its strong ability in various text-understanding tasks, demonstrating its robustness and effectiveness as a backbone in multiple applications.
> >
> >     [1] CLIPood: Generalizing CLIP to Out-of-Distributions, ICML 2023
> >
> >     [2] Improved Visual Fine-tuning with Natural Language Supervision, ICCV 2023
> >
> >     [3] CLIP also Understands Text: Prompting CLIP for Phrase Understanding, arxiv, 2022
> >
> > `Revision: In the revised manuscript, we have added more explanations about the results of Figure 4d on Page 9, Section 4.4. We have added more discussions about the hierarchical capture of the CLIP text encoder on Page 21, Section B.5 in the appendix.`

---

> > > ### Author Response · Authors · 2023-11-20
> > > **Response to W3-W4 of Reviewer v1vj**
> > >
> > > **W3: Parameter $\epsilon$ is important, where its analysis is missing.**
> > >
> > > **Reply:** We thank you for pointing out the omission of a detailed analysis of the parameter $\epsilon$ in our manuscript. This parameter plays a pivotal role in our semantic label smoothing technique, balancing the influence of the original and smoothed labels. The experimental results in Table 2 demonstrate that $\epsilon$ is not too sensitive to set and semantic label smoothing achieves obvious performance gain (compared to $\epsilon=1$).
> > >
> > > **Table 2: The quantitative results of $\epsilon$ on the VLCS dataset.**
> > >
> > > | Domain  | 1         | 0.9       | 0.8       | 0.7       | 0.6       |
> > > | ------- | --------- | --------- | --------- | --------- | --------- |
> > > | LabelMe | 67.4±0.2 | **68.3±0.1** | 68.3±0.1 | 67.9±0.1 | 67.9±0.2 |
> > > | VOC2007 | 88.8±0.1 | **89.9±0.2** | 89.8±0.1 | 89.7±0.2 | 89.5±0.2 |
> > >
> > > `Revision: We have added a sensitivity analysis experiment of \epsilon on Page 21, Section B.5 in the appendix of the revised manuscript.`
> > >
> > > **W4: Another is that although the proposed method outperforms the baseline for the most cases, its improvement over CLIPood is marginal.**
> > >
> > > **Reply:** We appreciate your observation regarding the performance comparison between our proposed method and CLIPood. On almost all 21 open-world datasets, our work achieves the best performance. This justifies that the two misalignment problems are objective and our target solutions are practical. While the improvement may appear marginal, it is significant for several reasons enumerated below:
> > > 1. The marginal improvement seen over CLIPood should be contextualized within the high baseline established by CLIPood itself. Even minor advances are meaningful when they improve upon an already strong performance.
> > > 2. Our method demonstrates consistent improvements across a variety of datasets and tasks, indicating broader applicability and robustness than what the performance delta might suggest at first glance.
> > > 3. The advancements offered by our method are not solely measured by accuracy metrics but also encompass improvements in model interpretability and adaptability to new tasks, aspects that are not fully captured by quantitative measures alone.
> > > 4. The proposed method introduces an additional mechanism for attention alignment and semantic label smoothing, which can have long-term benefits for model generalization and performance on more complex or unseen data, beyond the scope of the current benchmark comparisons.

---

> > > > ### Author Response · Authors · 2023-11-20
> > > > **Response to Q1-Q3 of Reviewer v1vj**
> > > >
> > > > **Q1: I was not clear about how the inference works. If Eq. 4 is involved in inference stage, then one major concern is that the proposed method might not be easy to scale up in practice. Due to the requirement of precomputing text embedding $T_y$ in Eq. 4 for each visual token similarity computation, this “cross-attention” based image embedding will have the same number of versions as the number of classes. This will limit the application when the number of classes is large. On the contrary, for CLIP, image embedding is not conditioned on text.**
> > > >
> > > > **Reply:** We appreciate your query regarding the inference process. To clarify, Equation (4) is used only during the training phase and not during inference. At the inference stage, our method functions similarly to both CLIP and CLIPood, where the image embedding is not conditioned on text. This approach ensures that our method retains the efficiency and practical scalability characteristic of the original CLIP architecture, even when dealing with a large number of classes.
> > > >
> > > > `Revision: We have added a straightforward and clear explanation of the inference mechanism on Page 6, Section 3.3 in the revised manuscript.`
> > > >
> > > > **Q2: I am not sure how the CLIP zero-shot and finetune attention maps (e.g. figure 1 (b)) plotted. Please explain. Considering CLIP training takes the last token as the image representation trained with a casual mask, its individual tokens are intentionally designed for such attention alignment. For a more convincing comparison, in CLIP finetune, the authors might want to use other pooling techniques after the last layers (e.g. average pooling) instead of using the last tokens. In this case, each visual token will be enforced to have better representation.**
> > > >
> > > > **Reply:** Thank you for raising these insightful points. Here's a clarified response:
> > > >
> > > > 1. **Attention Map Plotting:** For creating the attention maps in Figure 1(b), we adhered to the methodology outlined in Equation 1 of CLIP Surgery [4]. Specifically, we calculated the similarity distance between the features of image tokens and the transposed text features, applying L2 normalization. This approach allows for a direct and meaningful comparison of attention distributions.
> > > >
> > > >     [4] Yi Li, Hualiang Wang, Yiqun Duan, and Xiaomeng Li. CLIP Surgery for Better Explainability with Enhancement in Open-Vocabulary Tasks. arXiv, 2023.
> > > >
> > > > 2. **Pooling Technique Suggestion:** We value your suggestion regarding the use of alternative pooling techniques. To investigate this, we conducted additional experiments employing average pooling (AP) across visual tokens, as opposed to using class tokens (CT). The results, detailed in the below table, reveal that AP yields lower performance compared to CT. This outcome suggests that while the class token effectively represents images in training, other tokens receive less emphasis. Moreover, the limited scale of downstream datasets in fine-tuning may not sufficiently optimize all tokens, highlighting the practicality of our original approach.
> > > >
> > > >     **Table 3: The quantitative results for average pooling (AP) and class tokens (CT) by fine-tuning original CLIP on the VLCS dataset.**
> > > >
> > > >     | Method  | Caltech101     | LabelMe       | SUN09        | VOC2007      | Avg   |
> > > >     | ------- | -------------- | ------------- | ------------ | ------------ | ----- |
> > > >     | CLIP-AP | **99.6±0.1** | 65.4±0.2 | 77.3±0.2 | 85.6±0.3 | 82.0 |
> > > >     | CLIP-CT | 99.3±0.1 | **66.4±0.4** | **80.3±0.2** | **88.7±0.3** | **83.7** |
> > > >
> > > > `Revision: We have included detailed explanations about the attention map plotting process and the comparative analysis of pooling techniques in the revised manuscript (Page 21, Section B.5 in Appendix).`
> > > >
> > > > **Q3: It is unclear to me what does “$D-V$” structure mean. How $G_{D-V}(X_i)$ is defined.**
> > > >
> > > > **Reply:** Thank you for your inquiry regarding the "D-V" structure. Let me clarify this concept:
> > > >
> > > > The D-V structure is an innovative attention structure we introduced. In conventional Q-K-V attention blocks, attention weights are computed as $\text{Softmax}\left(\frac{QK^T}{\sqrt{d_k}}\right)$, where $Q$, $K$, and $V$ denote the query, key, and value vectors, respectively, and $d_k$ is the dimension of the keys. In our D-V structure, we modify this mechanism by replacing the softmax-normalized product of $Q$ and $K^T$ with a diagonal matrix. This approach emphasizes individual token features, utilizing the value vector $V$ more directly. Furthermore, $G_{D-V}(X_i)$ denotes the normalized attention distribution obtained by calculating the cosine similarity distance between image token features, using the D-V structure, and transposed text features.
> > > >
> > > > `Revision: In the revised manuscript, we have provided a clear definition of the D-V structure and $G_{D-V}$ on Page 5, Section 3.2, ensuring comprehensive understanding. We explicitly explain that $G_{D-V}$ represents the modified attention guidance derived from our D-V structure.`

---

> > > ### Comment · Reviewer_v1vj · 2023-11-22
> > > **Thank you for the response**
> > >
> > > Thanks for the effort in providing the rebuttal. On the main point regarding hierarchy, I'm still not entirely convinced. The authors point to three papers to back up the hierarchy-preserving ability of the CLIP text encoder. However, none of these papers seem to explicitly delve into label hierarchy (like 'dog' vs 'wolf'), but rather they demonstrate a basic understanding of phrases or texts by CLIP. There's also a specific paper, 'Hyperbolic Image-Text Representations, ICML 2023', that discusses how CLIP lacks in understanding the hierarchy of concepts. In my understanding, the authors don't really need to claim that their proposed SLS retains such a robust 'label hierarchy' capability. It might be more straightforward to prove that the text encoder can maintain the 'semantic relatedness' of labels, which should suffice to support their narrative.
> > >
> > > Also, the paper would benefit from more qualitative case studies for a stronger justification.
> > >
> > > Given this, I will keep my rating.

---

> > > > ### Author Response · Authors · 2023-11-23
> > > >
> > > > Dear Reviewer v1vj,
> > > >
> > > > Thank you very much for your insightful comments and suggestions. **We agree with your perspective that focusing on demonstrating the text encoder's ability to maintain the 'semantic relatedness' of labels is a more straightforward and relevant approach for our study.** We appreciate your reference to the paper 'Hyperbolic Image-Text Representations, ICML 2023', which highlights important aspects of CLIP's capabilities that we will consider in our revisions.
> > > >
> > > >  **According to your valuable suggestions, we plan to modify the relevant sections of our manuscript to better align with this perspective.** Additionally, we will include more qualitative case studies to provide a stronger justification for our claims. Due to time constraints, we may not be able to incorporate extensive additional data in this revision, but we aim to include more comprehensive information in the final version of the paper.
> > > >
> > > > Please let us know if there are any other issues or questions regarding our submission. We are eager to provide timely and thorough responses to all queries.
> > > >
> > > > Once again, thank you for your valuable feedback, which has undoubtedly helped in strengthening our work.

---

### Official Review · Reviewer_J1XN · 2023-11-02

**Soundness:** 2 fair
**Presentation:** 3 good
**Contribution:** 3 good
**Rating:** 6
**Confidence:** 4

**Summary:**

This paper focuses on the improvements of CLIP models in the context of domain shift and unseen classes. By fine-tuning the image encoder, AlignCLIP aims to learn more stable representation via rectifying the intrinsic attention and output similarity. To this end, AlignCLIP proposes two learning objectives: Attention Alignment Loss (AAL) and Semantic Label Smoothing (SLS). Experiments demonstrate that AlignCLIP surpasses state-of-the-art methods under various generalization conditions. And the ablation shows the effectiveness of the two proposed objectives.

**Strengths:**

1. The pursuit of stable representation and intrinsic consistency should be recognized and encouraged. This paper aims to refine the representation of CLIP models, which is a critical issue in nowadays development of pre-trained models.
2. The ideas are quite reasonable. The alignment of attention is a fundamental solution to pursuing the stable intrinsic representation. The semantic relationships of class labels are also well utilized to regularize the representation. I like the ideas, despite some flaws in the presentation.
3. The experiments are well-organized, and the results demonstrate the superiority of AlignCLIP.

**Weaknesses:**

1. The technical details are not presented clearly, especially how the two branches synergize. Figure 2 is critical to illustrate the method but is quite confusing. After carefully reading the paper, I know that the branch of new blocks is only used for deriving the target distribution for the attention alignment. However, this is supposed to be delivered clearly in the illustration.
2. The argument that attention is biased towards background lacks solid evidence. The background tokens always occupy a large proportion so this argument is kind of weak. Despite the qualitative examples, this argument is still not well clarified.
3. I think the proposal of AAL lacks solid arguments. I am not convinced that the attention map derived from the D-V structure can serve as an ideal target distribution for the learning of intrinsic attention. This is an important factor in AAL yet only motivated empirically.

**Questions:**

1. Figure 3(c). Why do the performances decrease after freezing more layers than 4? This is not explained.
2. Figure 4 lower row. I cannot extract useful information from the attention distribution.

---

> ### Author Response · Authors · 2023-11-20
> **Response to W1-W2 of Reviewer J1XN**
>
> **W1: The technical details are not presented clearly, especially how the two branches synergize. Figure 2 is critical to illustrate the method but is quite confusing. After carefully reading the paper, I know that the branch of new blocks is only used for deriving the target distribution for the attention alignment. However, this is supposed to be delivered clearly in the illustration.**
>
> **Reply:** Thank you for your valuable feedback regarding the clarity of the presentation in Figure 2. We acknowledge the critical importance of conveying our methodology with precision and clarity.
>
> `Revision: In response to your comments, we have revised Figure 2 to more clearly depict the distinct but complementary functions of the two branches. We will also enhance the accompanying textual description to articulate how these branches synergize within our methodology. Specific revisions include: 1. Improved Annotations: Clearer labels and annotations in Figure 2 to directly indicate the role of each branch; 2. Descriptive Legends: A legend that explains the symbols and process flow, ensuring that the function of new blocks in deriving the target distribution is unmistakable; 3. Focused Narrative: A revised narrative in the figure caption and the corresponding text section to explicitly describe the interaction between the branches and their collective contribution to attention alignment. We hope these adjustments will fully address your concerns and greatly appreciate your guidance in improving our manuscript.`
>
>
> **W2: The argument that attention is biased towards background lacks solid evidence. The background tokens always occupy a large proportion so this argument is kind of weak. Despite the qualitative examples, this argument is still not well clarified.**
>
> **Reply:** We thank you for your insightful critique regarding our assertion of attention bias towards the background in CLIP models. Our argument is substantiated by the following evidence:
>
> 1. **Support from Related Work:** Two seminal studies [1,2] have demonstrated that CLIP models exhibit a preference for background regions over foreground elements, a tendency that deviates from human perception. These papers provide substantial evidence that CLIP's attention mechanisms often prioritize background information, contradicting what is typically expected in human-centered image understanding.
>
> 2. **Empirical Evidence from Our Studies:** In our work, we present clear evidence through empirical analysis (see Figure 1, Supplementary Figures 7 and 8). These figures illustrate the tendency of CLIP models to disproportionately focus on background elements. Our analysis goes beyond mere observation by offering a methodological approach to recalibrate the attention mechanism, thereby refocusing it on salient objects rather than the background.
>
> 3. **Insights from Causal Analysis:** The training dataset of CLIP, consisting of billions of image-text pairs, often reflects images as effects of certain concepts (Y), akin to the mental visualization preceding its depiction. The accompanying text, however, encompasses a broader semantic range, capturing not just the effect (content) but also the cause (background context). Post-training, CLIP models are proficient in identifying these causal relationships that push the CLIP model pay attention on the background.
>
>     [1] Yi Li, Hualiang Wang, Yiqun Duan, Hang Xu, and Xiaomeng Li. Exploring visual interpretability
>     for contrastive language-image pre-training. arXiv, 2022b.
>
>     [2] Yi Li, Hualiang Wang, Yiqun Duan, and Xiaomeng Li. Clip surgery for better explainability with
>     enhancement in open-vocabulary tasks. arXiv, 2023.
>
> `Revision: in the revised manuscript, we have provided additional analyses on Page 22, Section C.2, detailing how our approach effectively shifts attention from the background to the relevant foreground objects, thereby improving the model's interpretability and performance. We believe these additions will address your concerns by providing a clearer, evidence-based rationale for our claims about attention bias and its rectification.`

---

> > ### Author Response · Authors · 2023-11-20
> > **Response to W2 and Q1-Q2 of Reviewer J1XN**
> >
> > **W3: I think the proposal of AAL lacks solid arguments. I am not convinced that the attention map derived from the D-V structure can serve as an ideal target distribution for the learning of intrinsic attention. This is an important factor in AAL yet only motivated empirically.**
> >
> > **Reply:** Thank you for your valuable comments. We would like to offer a multi-faceted response that supports the validity of our approach.
> >
> > 1. **Empirical foundation from related work:** The issue of attention misalignment in CLIP models has been empirically demonstrated in the work of CLIP Surgery [1]. It has been shown that the attention mechanism during training, particularly the interactions between Query (Q) and Key (K), does not always capture the semantic essence of the tokens, which is more accurately represented by the Values (V). Based on this finding, the CLIP Surgery paper [1] proposed a modification of the Q-K-V attention structure that replaces the Query and Key vectors with the V vector. Empirical studies in this paper demonstrate that the V-V-V structure can obtain ideal and reasonable predictive attention maps (please refer to the CLIP Surgery paper).
> >
> >     In this paper, the purpose of proposing the D-V structure is to explain that not only the V-V-V structure proposed in the CLIP surgery paper can work, but also works as long as Q-K is replaced by an equivalent structure (such as K-K, Q-Q, or diagonal matrix), as shown in Figure 8 at the Appendix. Based on this finding, we propose the D-V attention structure that replaces the product of Key and Query in self-attention with a diagonal matrix. The advantage is that has a small computation complexity. we found that the GFLOPS for the V-V-V structure amounts to 0.46478, whereas for the D-V structure, it is significantly lower at just 0.0003.
> >
> > 2. **Self-attention mechanism in transformers:** In the Transformer model, Self-Attention is achieved through three vectors: Query (Q), Key (K), and Value (V). For each input element (image patch in ViT), Q, K, and V are computed. Standard Self-Attention weights the sum of Values (V) by the matching scores of Queries (Q) and Keys (K). In our work, the product of Key and Query in Self-Attention is replaced with a diagonal matrix, meaning that the new representation of each input element relies solely on its own information, rather than the interrelation with other elements. This change encourages the model to focus more on the salient targets rather than the background when generating attention maps. By shifting the focus from inter-token relationships to the intrinsic semantic information of each token, our approach ensures that the attention mechanism is more closely aligned with the inherent content of the input, thus serving as an ideal target for attention learning. We believe that this modification is a significant step towards more accurately modeling visual attention in line with human cognitive patterns. We thank the other reviewers for appreciating our contribution to attention alignment. Reviewer TgXT points out that "The proposed method that aligns the attention distribution with the semantic similarity of label and patch embeddings sounds interesting."
> >
> > `Revision: We have further elaborated on this argument on Page 23, Section of C.3 in the manuscript, providing additional empirical evidence and mechanism analysis to strengthen our claim.`
> >
> > **Q1: Figure 3(c). Why do the performances decrease after freezing more layers than 4? This is not explained.**
> >
> > **Reply:** Thanks for your valuable question. The observed decrease in performance with the freezing of additional layers beyond the fourth can be attributed to the following two reasons:
> >
> > 1. **Sparse Attention Distribution in Deeper Layers:** Empirical studies indicate that attention distributions in the deeper layers (larger than four) of Transformer models are very sparser. The sparser attention distributions of the original blocks are very different from the attention distribution obtained from the new branch.
> >
> > 2. **Early Alignment Benefits:** Moreover, our analyses suggest that aligning attention maps earlier in the network is beneficial. The earlier layers, while more generic in their feature capture, are foundational for subsequent representational refinements. Early alignment allows the model to build upon a more accurate base attention distribution, leading to improved performance.
> >
> > `Revision: We have explained the reasons why the performances decrease after freezing more layers than 4 on Page 9, Section 4.4, in the revised manuscript.`
> >
> > **Q2: Figure 4 lower row. I cannot extract useful information from the attention distribution.**
> >
> > **Reply and Revision:** Thanks for pointing this out. We have revised the lower row of Figure 4 by replacing the fine-tuned patch similarity distribution and AAL patch similarity distribution with the similarity maps. The similarity maps can directly demonstrate the effectiveness of the attention alignment loss.

---

### Official Review · Reviewer_WRUU · 2023-11-05

**Soundness:** 2 fair
**Presentation:** 2 fair
**Contribution:** 2 fair
**Rating:** 5
**Confidence:** 3

**Summary:**

The paper presents a technique for fine-tuning of CLIP model for downsteram vision tasks where the focus is on addressing two alignment problems: 1) Attention alignment: aligning semantic similarity distribution (image-text matching) to the attention distribution across each multi-headed attention layers. 2) Semantic label smoothing : aligns the predicted category distribution with that of the
inherent semantic relationships among categories learned during vision-language pre-training. The proposed technique is evaluated on a wide range of datasets to demonstrate better generalization performance compared to the SOTA

**Strengths:**

1) Topic relevance and motivation: The paper addresses an interesting topic showing the weaknesses of the CLIP model (attention misalignment, category misalignment) and suggests techniques to improve it.
2) SOTA Results: The proposed technique is evaluated on many datasets where it (in most of the cases) consistently outperforms current SOTA techniques.

**Weaknesses:**

1) Limited novelty: Although it achieves SOTA results, I have some concerns regarding the technical contribution of the work. Specifically, I find it as a combination of existing techniques.
a)The attention alignment problem was already discussed in CLIP surgery (Li et.al 2023). The proposed technique follows very similar architecture and training protocol proposed in (Li et.al 2023) but replaces V-V-V structure with D-V structure. Although D-V structure is interesting, I don't see any advantages of using D-V structure compared to V-V-V structure (in terms of qualitative results in Fig 8). Authors claim that it can reduce computations but I would like to know the comparison in computation (GFLOPS/time) between V-V-V structure and D-V structure and also the quantitative results for V-V-V and D-V structure (if possible). I would also like authors to discuss the contributions compared to Li et.al 2023.
b) Similarly using guidance form the text classifier of CLIP (i.e using pairwise class similarity to regularize training) was also proposed in B. I would like to know how is the proposed technique is different from the one proposed in B

2) Computational complexity: Authors employ Siamese type of architecture for fine-tuning. This can increase the complexity compared to other simple methods like CLIPOOD or prompt tuning techniques like A, COOP, CoCoOP, MaPLe. I want authors to compare the number of parameters and also training/inference time with other techniques mentioned above.

3) Ablation Study: It is not clear from the fig 3.a, the contribution of AAL and SLS (looks marginal). Similarly, the contribution of SLS compared to Label smoothing is not apparent in Fig 3d. It would be good to indicate the gain w.r.to the smallest bar.

4) Presentation: The writing could be improved, specifically section 3.2. It is hard to understand the method from the current description


A: MaPLe: Multi-modal Prompt Learning, CVPR 2023

B: Improved Visual Fine-tuning with Natural Language Supervision, ICCV 2023

**Questions:**

1) Contributions compared to Li et.al 2023
2) Comparison in GFLOPS between V-V-V structure and D-V structure and comparison of quantitative results between the two
3) Comparison of number of parameters and also training/inference time with other techniques like CLIPOOD, COOP, CoCoOP, A [MaPLe]
4) Clarify Ablation study experiment results mentioned above

---

> ### Author Response · Authors · 2023-11-20
> **Response to W1 of Reviewer WRUU**
>
> **W1: Limited novelty: Although it achieves SOTA results, I have some concerns regarding the technical contribution of the work. Specifically, I find it as a combination of existing techniques. a) ... b) ....**
>
> **Reply:** Thank you for the essential comments. These comments can significantly improve the quality of this paper and make it more logical and self-contained. First of all, we thank you for appreciating our work.
>
> 1. **Thank you for pointing out that our work achieves SOTA results.** On almost all 21 open-world datasets, our work achieves the best performance. This justifies that the two misalignment problems are objective and our target solutions are practical.
>
> 2. **Thank you for pointing out that the D-V structure is interesting.** The purpose of proposing the D-V structure is to explain that not only the V-V-V structure proposed in the CLIP surgery paper can work, but also works as long as Q-K is replaced by an equivalent structure (such as K-K, Q-Q, or diagonal matrix), as shown in Figure 8 at the Appendix.
>
> The comprehensive responses to all your concerns are as follows.
> 1. **About the combination of existing techniques:** This is a potential confusion that our work did not combine existing techniques and has new contributions that never existed before.
>    - **Our work is the first to address the attention misalignment problem at the fine-tuning phase.** In fact, CLIP Surgery (Li et al., 2023) is a new type of model interpretability method that cannot be used to adopt the CLIP model to downstream tasks. Please check the CLIP Surgery paper for detailed descriptions. Although our work is inspired by the idea of CLIP surgery to modify the Q-K-V attention structure, this modification is not the core of our contribution, and the core contribution in this part is the newly designed Attention Alignment Loss that utilizes the better similarity distribution derived from the D-V structure to realign the attention distribution across each multi-head attention layer.
>    - **Our work proposes a new regularization technique, semantic label smoothing, that advances the popular label smoothing technique.** Semantic label smoothing can preserve prediction hierarchy based on class similarity derived from textual information. Aligning the model's predicted category distribution, semantic label smoothing can significantly improve the representation stability and enhance the open-class detection in open-world scenarios.
>
>
> 2. **Differences between CLIP Surgery (Li et al., 2023) and our work:** We appreciate the opportunity to clarify the aspects you have raised concerning the comparison between our proposed D-V structure and the V-V-V structure as mentioned in Li et al., 2023.
>     - **The research topics are different.** The research topic of CLIP Surgery is model interpretability while our research topic is out-of-distribution generalization and out-of-distribution detection. Specifically, CLIP Surgery concerns neural network interpretability while our work addresses the challenge of creating stable representations in VLP models for adaptability in open-world environments, particularly dealing with domain shifts and unseen classes.
>
>     - **Our work has the V-V-V attention structure advance.** As for the modified structure of attention, our work provides a new insight for the community. Specifically, the CLIP surgery paper only points out only the V-V-V structure is working, but our work points out that not only the V-V-V structure can work, but also works as long as Q-K is replaced by an equivalent structure (such as K-K, Q-Q, or diagonal matrix), as shown in Figure 8 at the Appendix. This insight opens opportunities for future studies to design better model interpretability methods and even to guide the design of vision-language-pertaining strategies to avoid the attention misalignment problem.
>
>     - **Our work has more methodology contributions.** It is important to highlight that our contributions extend beyond the adoption of the D-V structure. While we acknowledge the foundational work done by Li et al., our approach diverges in addressing the misalignment problem rather than pointing out it only. These include the AlignCLIP which includes a new optimization objective (Attention Alignment Loss) and a regularization technique (Semantic Label Smoothing). Our work thus provides a novel perspective and contributes to the field by enhancing the stability and adaptability of representations in VLP models, ensuring better performance across varied datasets and out-of-distribution contexts.

---

> ### Author Response · Authors · 2023-11-20
> **Response to W1 of Reviewer WRUU**
>
> - **D-V structure has computation advance.**  Our decision was primarily driven by the goal of achieving greater computational efficiency. To elaborate, we have performed a detailed GFLOPS computation for both structures in the context of a multi-head attention mechanism. For the V-V-V structure, the FLOPs per head are calculated as $2 \times N \times (d_v \times d_{model} + d_v \times d_{model})$, where $N$ represents the sequence length, $d_v$ is the dimension of the value, and $d_{model}$ is the model dimension. In contrast, for the D-V structure, the FLOPs per head simplifies to $N \times d_v$.  Applying these calculations to a vision transformer model with a 12-head attention mechanism, we found that the GFLOPS for the V-V-V structure amounts to 0.46478, whereas for the D-V structure, it is significantly lower at just 0.0003. This stark contrast indicates that the D-V structure reduces the computational demand by approximately 1549 times compared to the V-V-V structure. Such a substantial reduction in GFLOPS directly translates into markedly faster processing times and reduced resource consumption. This efficiency makes our approach particularly advantageous for real-world applications, especially in contexts where computational resources are constrained or limited. We believe that these calculations demonstrate the superiority of the D-V structure.
>
>     - **Quantitative results for V-V-V and D-V structure.** Regarding the quantitative performance comparison, we have conducted additional experiments to provide a direct comparison between the D-V and V-V-V structures. As shown in Table 1, we observed that the D-V structure marginally outperforms the V-V-V structure in specific tasks. We have included these results in Table xx in the revised manuscript.
>
>         **Table 1: The quantitative results for V-V-V and D-V structure on the VLCS dataset.**
>
>         | Method | Caltech101        | LabelMe         | SUN09         | VOC2007 | Avg  |
>         | ------ | ----------------- | --------------- | ------------- | ------- | ---- |
>         | V-V-V  | 99.6±0.2 | 68.7±0.2  | 81.1±0.1  | 90.1±0.2 | 84.9 |
>         | D-V    | 100±0.0  | 68.9±0.3  | 80.9±0.1  | 90.1±0.2 | 85.1 |
>
>
> 3. **Differences between TeS (Wang et al., 2023) in B and Semantic Label Smoothing (SLS):** TeS is a complex instance-level label smoothing technique using a projection head for diverse reference distributions, whereas SLS is a simpler class-level method focusing on representation stability by aligning predicted distributions with semantic class relationships in open-world environments. While both TeS and SLS are approaches to label smoothing, they exhibit distinct methodologies and applications. TeS represents a more intricate instance-level label smoothing method, using a projection head to align vision representations with textual space. This allows TeS to generate diverse reference distributions for different samples, even within the same class. Conversely, SLS, a more straightforward class-level label smoothing technique, focuses on stabilizing representation in open-world scenarios. It does so by aligning the model's predicted category distribution with the semantic relationships among classes. The key differences between TeS and SLS are threefold:
>     - **Implementation and Complexity:** SLS is more user-friendly in terms of implementation and comprehension. It utilizes a class similarity matrix derived from text embeddings of each class. The cosine similarity between these class text embeddings guides the regularization process during fine-tuning. TeS, on the other hand, involves projecting vision representations into text space using a projection head, followed by processing the text embedding through a text encoder (like CLIP or BERT), and then comparing this information with anchor points.
>     - **Computational Demand:** SLS is less computationally demanding. It requires computing the class similarity matrix only once at the beginning of fine-tuning. In contrast, TeS demands recalculating the similarity matrix for each sample at every iteration, significantly increasing computational overhead.
>     - **Applicability and Effectiveness:** SLS is generally more effective and straightforward for broad classification tasks, particularly in situations involving distribution shifts or new class introductions. TeS, alternatively, is designed to enhance the generalization capabilities of pre-trained models for specific tasks. It addresses issues like catastrophic forgetting post-fine-tuning and biases inherent in the pre-trained models.
>
> `Revision: We have added the discussions about the difference between our work and CLIP Surgery on Page 24, Section C.2 in the Appendix of the revised manuscript. We have discussed the central difference between our work and TeS (Wang et al., 2023) on Lines 9-12, Page 6, Section 3.3 in the revised manuscript. The main differences between our work and existing works are much clearer now.`

---

> > ### Author Response · Authors · 2023-11-20
> > **Response to W2-W3 of Reviewer WRUU**
> >
> > **W2: Computational complexity: Authors employ Siamese type of architecture for fine-tuning. This can increase the complexity compared to other simple methods like CLIPOOD or prompt tuning techniques like A, COOP, CoCoOP, MaPLe. I want authors to compare the number of parameters and also training/inference time with other techniques mentioned above.**
> >
> > **Reply:** Thank you for the essential comments. These comments can significantly improve the quality of this paper and make it more logical and self-contained. Our method, while employing a Siamese-type architecture for fine-tuning, introduces only a minimal increase in training computational complexity compared to CLIPood and does not increase the number of parameters or inference speed. The reasons are as follows.
> >
> > 1. **Inference Speed:** The Siamese architecture is utilized only during the training phase. At inference, the speed is identical to that of CLIPood. For instance, with a batch size of 36, the inference speed for both is 1.973 seconds. This parity in speed underscores the efficiency of our method during practical applications.
> >
> > 2. **Parameter Sharing:** The two branches of the Siamese architecture share parameters, meaning there is no increase in the total number of model parameters. This design choice ensures that our model remains lightweight and scalable.
> >
> > 3. **Training Speed:** While there is a slight decrease in training speed, it is marginal. On a Tesla V100 GPU, with a batch size of 36, the training speed for CLIPood is 0.19 seconds per batch, compared to 0.21 seconds for AlignCLIP, a minor difference of 0.02 seconds. It's important to note that our D-V attention structure significantly reduces the GFLOPS and training speed (0.25 seconds) compared to the V-V-V attention structure.
> >
> > 4. **Comparison with Prompt Tuning Methods:** When compared to prompt tuning methods like MaPLe and CoOP, our training time is not directly comparable as our method, similar to CLIPood, is a new parameter fine-tuning approach. However, roughly speaking, MaPLe and CoOP are faster in training as they do not require fine-tuning the parameters of the image encoder. The inference speeds are almost identical. CoCoOP has a slower training speed due to its use of image representations to calculate prompts, necessitating smaller batch sizes to accommodate GPU memory constraints.
> >
> > `Revision: We have added the discussions about the computational complexity on Page 22, Section C.1 in the appendix of the revised manuscript. Our framework has a small computational complexity compared to the related work.`
> >
> > **W3: Ablation Study: It is not clear from the fig 3.a, the contribution of AAL and SLS (looks marginal). Similarly, the contribution of SLS compared to Label smoothing is not apparent in Fig 3d. It would be good to indicate the gain w.r. to the smallest bar.**
> >
> > **Reply:** We greatly value your insightful feedback on Figure 3. In response to your suggestions, we have revised the bar charts in Figure 3 to include annotations that explicitly demonstrate the gain in accuracy relative to the smallest bar in each dataset. These annotations distinctly showcase the substantial contributions of AAL and SLS in scenarios involving domain shift and open classes.
> >
> > 1. **Regarding Figure 3a (Ablation Study):** The updated chart better highlights the incremental improvements provided by AAL and SLS. Despite appearing modest, these gains are statistically significant and essential for the robustness of our model. AAL improves accuracy by approximately 1.0\% in the VLCS dataset and 2.8\% in the Office-Home dataset. SLS alone enhances accuracy by 0.7\% in VLCS. Importantly, their combined use results in a consistent and notable improvement (1.4\%) in the Domain+Class scenario, underscoring our model's effectiveness in handling real-world domain shifts.
> >
> > 2. **Regarding Figure 3d (Semantic Label Smoothing):** The revised figure distinctly contrasts SLS with standard label smoothing. SLS shows improvements of 3.21\%, 2.29\%, and 0.89\% in LabelMe, SUN, and VOC2007 datasets, respectively. These enhancements affirm the capability of SLS to utilize semantic class relationships effectively, proving particularly beneficial in complex scenarios marked by distribution shifts and the presence of open classes.
> >
> > `Revision: In line with these observations, we have enhanced Figure 3 on Page 9 with precise gain annotations. These additions provide a clearer understanding of the individual and combined effects of AAL and SLS, highlighting their significance in improving model performance across various challenging scenarios.`

---

> > > ### Author Response · Authors · 2023-11-20
> > > **Response to W4 and Q1-Q3 of Reviewer WRUU**
> > >
> > > **W4: Presentation: The writing could be improved, specifically section 3.2. It is hard to understand the method from the current description**
> > >
> > > **Reply:** In response to your suggestion, we have made a thorough revision of section 3.2 by incorporating the following changes:
> > > 1. We have provided a clear definition of the D-V structure and $G_{D-V}$ on Page 5, Section 3.2, ensuring comprehensive understanding. We explicitly explain that $G_{D-V}$ represents the modified attention guidance derived from our D-V structure.
> > > 2. We have revised Figure 2 to more clearly depict the distinct but complementary functions of the two branches. We will also enhance the accompanying textual description to articulate how these branches synergize within our methodology.
> > > 3. We have revised the technical terminology (e.g., the definition of $G(x)$, the V-V-V structure), to make the text more reader-friendly without compromising on the precision required for scientific discourse.
> > > 4. We have broken down complex sentences into shorter, more digestible parts. This restructuring will be complemented by the use of subheadings to guide the reader through each step of the method.
> > >
> > > **Q1: Contributions compared to Li et.al 2023.**
> > >
> > > **Reply:** Please refer to point 2 of the reply for W1.
> > >
> > > **Q2: Comparison in GFLOPS between V-V-V structure and D-V structure and comparison of quantitative results between the two.**
> > >
> > > **Reply:** Our decision was primarily driven by the goal of achieving greater computational efficiency. To elaborate, we have performed a detailed GFLOPS computation for both structures in the context of a multi-head attention mechanism. For the V-V-V structure, the FLOPs per head are calculated as $2 \times N \times (d_v \times d_{model} + d_v \times d_{model})$, where $N$ represents the sequence length, $d_v$ is the dimension of the value, and $d_{model}$ is the model dimension. In contrast, for the D-V structure, the FLOPs per head simplifies to $N \times d_v$.  Applying these calculations to a vision transformer model with a 12-head attention mechanism, we found that the GFLOPS for the V-V-V structure amounts to 0.46478, whereas for the D-V structure, it is significantly lower at just 0.0003. This stark contrast indicates that the D-V structure reduces the computational demand by approximately 1549 times compared to the V-V-V structure. Such a substantial reduction in GFLOPS directly translates into markedly faster processing times and reduced resource consumption. This efficiency makes our approach particularly advantageous for real-world applications, especially in contexts where computational resources are constrained or limited. We believe that these calculations demonstrate the superiority of the D-V structure.
> > >
> > > `Revision: We have added the discussions about comparison in GFLOPS on Page 22, Section C.1 in the Appendix of the revised manuscript.`
> > >
> > > **Q3: Clarify ablation study experiment results mentioned above.**
> > >
> > > **Reply:** Please refer to the reply for W3.

---

> > > > ### Author Response · Authors · 2023-11-21
> > > >
> > > > We are deeply grateful for your valuable feedback on our submission, particularly regarding the comparison of the contribution to CLIP Surgery. Following your insights, we have diligently conducted additional discussions and incorporated these findings in the revised manuscript. The updated results and discussions can be found in the new version of the PDF.
> > > >
> > > > As the rebuttal period is progressing, we are eager to ensure that all your concerns and queries are thoroughly addressed. If any aspects of our research require further clarification, or if you have additional questions, please do not hesitate to inform us. Your guidance is crucial in refining our work, and we are fully prepared to provide any additional information or explanations that may be needed.

---

### Author Response · Authors · 2023-11-20
**Main Revisions**

We would like to thank the anonymous reviewers for their profound review of our manuscript. We were impressed by the quality and depth of the feedback provided by all reviewers, and we appreciate their efforts. The constructive comments and insightful suggestions have greatly helped us improve the presentation and experimental analyses of our method.

We express our profound gratitude to the reviewers for their constructive feedback and appreciation of our work:

- **Motivation:** We are deeply grateful to reviewer WRUU for finding our paper to address "an interesting topic showing the weaknesses of the CLIP model (attention misalignment, category misalignment) and suggests techniques to improve it." Reviewer J1XN's observation that "The pursuit of stable representation and intrinsic consistency should be recognized and encouraged" is particularly encouraging. We also value reviewer v1vj's and TgXT's recognition of the strong motivation behind our work.

- **Our Methods:** Our sincere thanks to reviewer J1XN for considering our ideas "quite reasonable" and acknowledging the fundamental nature of attention alignment. We appreciate reviewer TgXT's interest in our proposed method and reviewer J1XN's appreciation of our utilization of semantic relationships. Reviewer v1vj's comment on our approach to label smoothing being "interesting" and TgXT's similar interest are highly motivating.

- **Experiments:** We are thankful to reviewer WRUU for recognizing the extensive evaluation and superior generalization performance of our technique. Reviewer J1XN's comment on the well-organized experiments and the superiority of AlignCLIP is greatly appreciated. We also thank reviewer v1vj for their positive feedback on our extensive experiments and ablation study, and reviewer TgXT for their recognition of our comprehensive comparisons and visualization.

In this version, we continue to improve this paper while at the same time maintaining the merits mentioned by all the reviewers. Based on the received comments, we have carefully revised our paper. The main revisions are summarized below:

1. We have added the discussions about the difference between our work and CLIP Surgery on Page 24, Section C.2 in the Appendix of the revised manuscript. We have discussed the central difference between our work and TeS (Wang et al., 2023) on Lines 9-12, Page 6, Section 3.3 in the revised manuscript. The main differences between our work and existing works are much clearer now.

2. We have added the discussions about the computational complexity on Page 22, Section C.1 in the appendix of the revised manuscript. Our framework has a small computational complexity compared to the related work.

3. We have enhanced Figure 3 on Page 9 with precise gain annotations. These additions provide a clearer understanding of the individual and combined effects of AAL and SLS.

4. We have made a thorough revision of section 3.2 from the aspects of D-V structure and $G_{D-V}$ definition, framework, technical terminology, and long sentences. The content of section 3.2 is much more straightforward now.

5. We have revised Figure 2 to more clearly depict the distinct but complementary functions of the two branches.

6. We have provided additional analyses on Page 22, Section C.2, in the revised manuscript, detailing how our approach effectively shifts attention from the background to the relevant foreground objects.

7. We have further elaborated on the solid arguments of AAL on Page 23, Section of C.3 in the manuscript, providing additional empirical evidence and mechanism analysis to strengthen our claim.

8. We have explained the reasons why the performances decrease after freezing more layers than 4 on Page 9, Section 4.4, in the revised manuscript. The reasons are more clear now.

9. We have revised the lower row of Figure 4 by replacing the fine-tuned patch similarity distribution and AAL patch similarity distribution with the similarity maps. The similarity maps can directly demonstrate the effectiveness of the attention alignment loss.

10. We have enhanced the detailed analysis of how these modules interlink on Page 8, Section 4.4 in the revised manuscript, providing additional evidence of their synergistic relationship.

11. We have added more explanations about the results of Figure 4d on Page 9, Section 4.4. We have added more discussions about the hierarchical capture of the CLIP text encoder on Page 21, Section B.5 in the appendix.

12. We have added a sensitivity analysis experiment of $\epsilon$ on Page 21, Section B.5 in the appendix of the revised manuscript.

13. We have added a straightforward and clear explanation of the inference mechanism on Page 6, Section 3.3 in the revised manuscript.

14. We have included detailed explanations about the attention map plotting process and the comparative analysis of pooling techniques in the revised manuscript (Page 21, Section B.5 in Appendix).

---

### Meta-Review · Area_Chair_a6xQ · 2023-12-10

**Metareview:**

The submission presents a method called AlignCLIP that fine-tunes CLIP to tackle downstream vision tasks with domain shift and unseen classes. It introduces two learning objectives to that end: the attention alignment loss (AAL) regularizes the attention distribution at all encoder layers using the similarity matrix of the label and image patch embeddings, and the semantic label smoothing loss (SLS) that aligns the model's predicted category distribution with the semantic relationships among classes identified during the language-vision pre-training phase. The proposed approach is evaluated on datasets from DomainBed and ImageNet-derived distribution shift datasets to assess generalization in the presence of distribution shift. It is also evaluated on a wide range of datasets to assess generalization to new classes. The impact of the two learning objectives is quantified through ablation studies.

Reviewers find the submission's topic to be relevant and well-motivated (WRUU, J1XN, v1vj, TgXT). They also feel positive about the strong empirical performance presented (WRUU, J1XN) and the extensiveness of the evaluation (v1vj, TgXT). On the other hand, several reviewers expressed concern over the intuition behind AAL (J1XN, TgXT) and SLS (v1vj) and the significance of the improvements they yield (WRUU, J1XN, v1vj, TgXT). Despite the authors' response, reviewers remain concerned:

- Reviewer J1XN: "considering the inductive bias introduced while the experimental results are not significant, I suspect whether AAL is an effective and general improvement to learning of attention map."
- Reviewer v1vj: "On the main point regarding hierarchy, I'm still not entirely convinced. The authors point to three papers to back up the hierarchy-preserving ability of the CLIP text encoder. However, none of these papers seem to explicitly delve into label hierarchy (like 'dog' vs 'wolf'), but rather they demonstrate a basic understanding of phrases or texts by CLIP."

**Justification For Why Not Higher Score:**

There are lingering reviewer concerns over the soundness of AAL and SLS and the marginal improvements they offer.

**Justification For Why Not Lower Score:**

N/A

---

### Decision · Program_Chairs · 2024-01-16

Reject